**Minimum forest cover required for sustainable water flow regulation of a watershed: a**
**case study in Jambi Province, Indonesia."**
Suria Tarigan[1], Kerstin Wiegand[2], Sunarti[3], Bejo Slamet[4]
[1]Department of Soil Sciences and Natural Resource Management, Bogor Agricultural
University, Indonesia
[2]Department of Ecosystem Modeling, University of Göttingen, Büsgenweg 4, 37077
Göttingen, Germany
[3]Faculty of Agriculture, University of Jambi, Jambi, Indonesia
[4]Faculty of Agriculture, North Sumatra University, Medan, Indonesia
Correspondence to: Suria Tarigan (E-mail: sdtarigan@apps.ipb.ac.id)

**Abstract**

In many tropical regions, the rapid expansion of monoculture plantations has led to a sharp
decline in forest cover, potentially degrading the ability of watersheds to regulate water flow.
Therefore, regional planners need to determine the minimum proportion of forest cover that is
required to support adequate ecosystem services in these watersheds. However, to date, there
has been little research on this issue, particularly in tropical areas where monoculture
plantations are expanding at an alarming rate. Therefore, in this study, we investigated the
influence of forest cover and oil palm (*Elaeis guineensis*) and rubber (*Hevea brasiliensis*)
plantations on the partitioning of rainfall into direct runoff and subsurface flow in a humid,
tropical watershed in Jambi Province, Indonesia. To do this, we simulated streamflow with a
calibrated Soil and Water Assessment Tool (SWAT) model and observed several watersheds
to derive the direct runoff coefficient (C) and baseflow index (BFI). The model had a strong
performance, with Nash–Sutcliffe efficiency values of 0.80–0.88 (calibration) and 0.80–0.85
(validation), and percent bias values of $-2.9$ to 1.2 (calibration) and 7.0–11.9 (validation). We
found that the percentage of forest cover in a watershed was significantly negatively
correlated with C and significantly positively correlated with BFI, whereas the rubber and oil
palm plantation cover showed the opposite pattern. Our findings also suggested that at least
30 % of the forest cover was required in the study area for sustainable ecosystem services.
This study provides new adjusted crop parameter values for monoculture plantations,

particularly those that control surface runoff and baseflow processes, and also describes the

quantitative association between forest cover and flow indicators in a watershed, which will

help regional planners in determining the minimum proportion of forest and the maximum

proportion of plantation to ensure that a watershed can provide adequate ecosystem services.

## 1   Introduction

In recent years, monoculture plantations have rapidly expanded in Southeast Asia, and the

areas under oil palm (*Elaeis guineensis*) and rubber (*Hevea brasiliensis*) plantations are

expected to further increase (Fox et al., 2012; Van der Laan et al., 2016). In Indonesia, which

is currently the largest palm oil producer worldwide, the oil palm plantation area increased

from 7 000 $km^2$ in 1990 to 110 000 $km^2$ in 2015 (Ditjenbun, 2015; Tarigan et al., 2016b), and

a further 170 000 - 200 000 $km^2$ are projected for future oil palm development (Colchester et

al., 2006; Wicke et al., 2011; Afriyanti et al., 2016). This rapid expansion of oil palm

plantations has been partly triggered by an increased demand for biofuel production

(Mukherjee and Sovacoo, 2014). In addition, rubber plantations, which are also prevalent in

Southeast Asia (Ziegler et al., 2009), currently cover 35 000 $km^2$ of land in Indonesia

(Ditjenbun, 2015).

Although oil palm is of economic value to farmers and the local regions in which it is

grown, it has received environmental and social criticism, often being held responsible for

deforestation (Wicke et al., 2011; Vijay et al., 2016; Gatto et al., 2017), biodiversity loss

(Fitzherbert et al., 2008; Koh and Wilcove, 2008; Wilcove and Koh, 2010; Carlson et al.,
2012; Krashevska et al., 2015), decreased soil carbon stocks (Guillaume et al., 2015, 2016;
Pransiska et al., 2016), and increased greenhouse gas emissions (Allen et al., 2015; Hassler et
al., 2017). Similarly, rubber plantations have environmental impacts such as reducing the soil
infiltration capacity, accelerating soil erosion, increasing stream sediment loads (Ziegler et
al., 2009; Tarigan et al., 2016b), and decreasing soil carbon stocks (Ziegler et al., 2011).
Furthermore, the conversion of tropical rainforest into oil palm and rubber plantations affects
the local hydrological cycle by increasing transpiration (Ziegler et al., 2009; Sterling et al.,
2012; Röll et al., 2015; Hardanto et al., 2017), increasing evapotranspiration (ET) (Meijide et
al., 2017), decreasing infiltration (Banabas et al., 2008; Tarigan et al., 2016b), increasing the
flooding frequency (Tarigan, 2016a), and decreasing low flow levels (Yusop et al., 2007;
Adnan and Atkinson, 2011; Comte et al., 2012; Merten et al., 2016). These climatic impacts
that occur due to land use change are expected to be stronger under maritime conditions, such
as those in Indonesia, than under continental conditions because 40 % of the global tropical
latent heating of the upper troposphere occurs over the maritime continent (Van der Molen et
al., 2006).

The forests in Jambi Province, Indonesia have been largely transformed into plantations

(Drescher et al., 2016), resulting in inhabitants experiencing water shortages during the dry
season and a dramatic increase in flooding frequency during the wet season (Merten et al.,
2016; Tarigan, 2016a) because plantations promote higher levels of direct runoff than
forested lands (Bruijnzeel, 1989, 2004; Tarigan et al., 2016b; Dislich et al., 2017). However,
this negative impact of plantation expansion could be minimized by maintaining an adequate
proportion of forested land as a watershed, which raises the question, what is the minimum
proportion of forest cover that is required in a watershed to support adequate water flow
regulation?
The water flow regulation function of watersheds represents their ability to retain rain
water and is one of the most important soil hydrological processes in tropical regions where
rainfall is highly seasonal (Lele, 2009). Functional water flow regulation by a watershed
reduces flood peaks by moderating direct runoffs (Le Maitre et al., 2014; Ellison et. al, 2017)
via soil water infiltration through the soil surface and percolation through the soil profile.
This vertical movement of water through the soil determines how much water flows as direct
runoffs and how much reaches the water table where it sustained as baseflow or groundwater
(Hewlett and Hibbert, 1967; Bruijnzeel, 1990; Le Maitre et al., 2014; Tarigan et al., 2016b).
Forest vegetation provides organic matter and habitat for soil organisms, thereby facilitating
higher levels of infiltration than other land uses (Hewlett and Hibbert, 1967).
A number of empirically based and process-based approaches can be used for assessing
the impacts of expanding rubber and oil palm plantations on hydrological characteristics in
the Southeast Asia region. Empirically based approaches use long-term historical data to

correlate land use changes with corresponding streamflow data (Adnan and Atkinson, 2011; Rientjes et al., 2011; Mwangi et al., 2016) or paired catchment studies (Bosch and Hewlett, 1982; Brown et al., 2005), whereas process-based approaches use physically based hydrological models in which the impact of land use changes is determined by varying the land use/cover settings (Khoi and Suetsugi, 2014; Guo et al., 2016; Zhang et al., 2016; Marhaento et al., 2017; Wangpimool et al., 2017). Process-based approaches have the drawback of requiring more data to be input and having high uncertainty in parameter estimation (Xu et al., 2014; Zhang et al., 2016). However, there is currently an absence of long-term historical data for Jambi Province, precluding the use of an empirically based approach.

Distributed hydrological models are useful for understanding the effects of land use changes on watershed flow regulation. Once such model is the Soil and Water Assessment Tool (SWAT) ecohydrological model (2012), which quantifies the water balance of a watershed on a daily basis (Neitsch et al., 2011) and has been recommended for evaluating the hydrological ecosystem services of a watershed (Vigerstol and Aukema, 2011). The SWAT model approach is one of the most widely used and scientifically accepted tools for assessing water management in a watershed (Gassman et al., 2007). Consequently, its popularity has also increased in Southeast Asia. Marhaento et al. (2017) recently used the SWAT model to analyze the impact of forest cover and agriculture land use on the runoff

coefficient (C) and baseflow index (BFI) on Java Island, Indonesia and found that a decrease

in forest cover from 48.7 % to 16.9 % and an increase in agriculture area from 39.2 % to

45.4 %, increased C from 35.7 % to 44.6 % and decreased BFI from 40 % to 31.1 %.

Meanwhile, Wangpimool et al. (2017) found that the expansion of rubber plantations in

Thailand between 2002 and 2009 led to an annual reduction of approximately 3 % in the

average water yield of the basin, whereas Babel et al. (2011) found that the expansion of oil

palm plantations in Thailand increased nitrate loading (1.3 %–51.7 %) in the surface water

based on SWAT simulations. Tarigan et al. (2016b) also used the SWAT model to simulate

the impact of soil and water conservation practices on low flow levels in oil palm-dominated

watersheds in Jambi Province, Indonesia. This study aimed to quantify the minimum

proportion of forest cover that is required to allow a watershed to provide adequate ecosystem

services. We selected Jambi Province as our study area because of the rapid expansion of oil

palm and rubber plantations in that area. The study findings provide new adjusted values for

crop parameters of monoculture plantations, particularly those that control surface runoff and

baseflow processes, and describe the quantitative association between forest cover and flow

indicators in a watershed, which will help regional planners in determining the minimum

proportion of forest that needs to be conserved to ensure that a watershed can provide

adequate ecosystem services.

## 2  Methods

### 2.1 Study area

The study area is located in Jambi Province, Sumatra (1° 54′ 31.4″ S, 103° 16′ 7.9″ E, Fig. 1). There has been a rapid expansion of plantations in this area, particularly oil palm and rubber (Drescher et al., 2016). The area has a tropical humid climate, with an average temperature of 27 °C and an average rainfall of 2700 mm year$^{-1}$. The rainy season occurs from October to March. The area under oil palm plantation in the study area (Jambi Province) increased from 1 500 km$^2$ in 1996 to 6 000 km$^2$ in 2011, representing an almost 400 % increase (Setiadi et al., 2011), whereas the area under rubber plantation increased from 5 000 to 6 500 km$^2$ over the same period (Ditjenbun, 2015). In 2013, only 30 % of Jambi Province was covered with rainforest (mainly located in mountainous regions), with 55 % of the land having been converted into agricultural land, of which 10 % was degraded/fallow and will potentially be converted into monoculture plantations (Drescher et al., 2016).

The study area consists of two macro watersheds for the simulation of the C and BFI values with the SWAT model, namely Batanghari Hulu (BH; Fig. 1a) and Merangin Tembesi (MT; Fig. 1b) watersheds, which cover areas of 18 415 km$^2$ and 13 452 km$^2$, respectively. The dominant land uses in both watersheds are forest (BH, 50 %; MT, 30 %) and plantation (BH, 18 %; MT, 48 %). The dominate soil types in the study area are classified as Tropodult

and Dystropept, which are characterized as consisting of medium to heavy texture (Allen et
al., 2015).
To ensure that the C and BFI values obtained from the macro watershed simulations
(particularly those sub-watersheds that were dominated by oil palm or rubber) reflected the
real observed values in the field, we carried out the direct C measurements in two small
watersheds having size 14 ha and 9 ha respectively in the study area (Fig. 1c). These small
watersheds were covered with 90 % oil palm and 80 % rubber plantations, respectively.

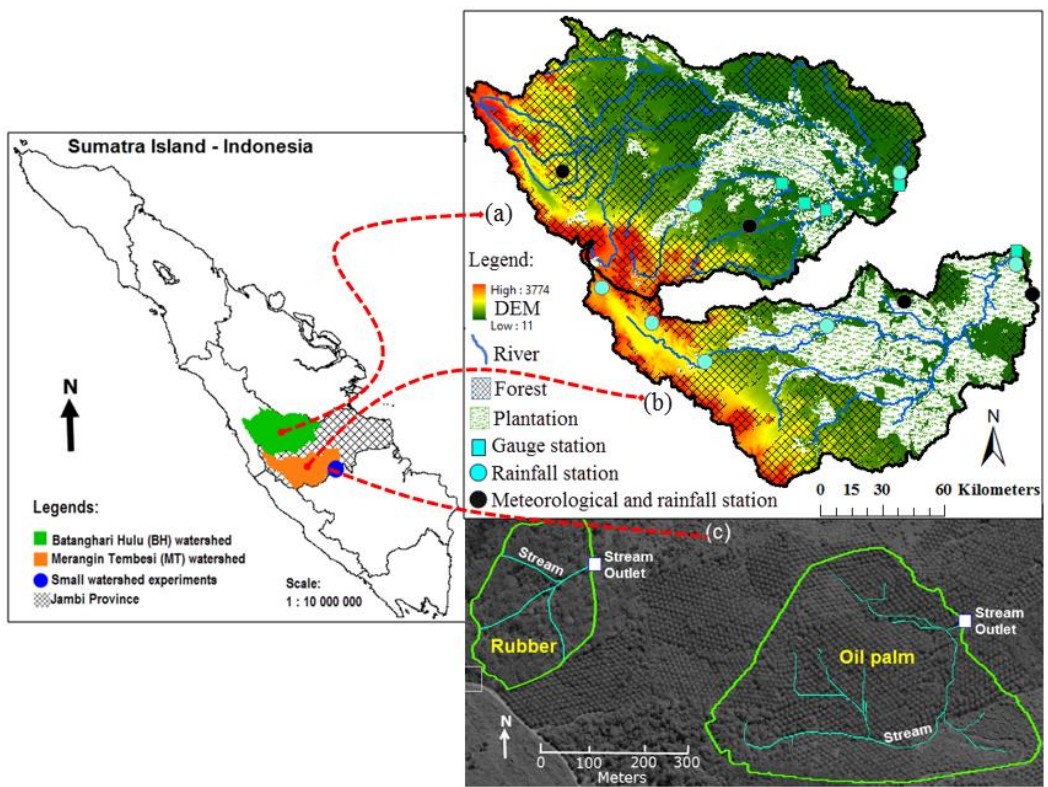


Figure 1. Locations of the (a, b) macro and (c) small watershed experiments in Jambi
Province, Sumatra, Indonesia.
Oil palm and rubber are perennial crops that have a life cycle of 25 years. Both crops are
planted in rows at planting distances of 8 and 4 m, respectively. In oil palm plantations, there
are two types of paths between the planting rows: the harvest path, which is used to transport
freshly harvested fruit bunches, and the so-called death path, which is used for piling pruned
leaf fronds, which occupy approximately 2 m or one-quarter of this path. Both oil palm and
rubber require very intensive harvesting activities, which occur twice per month for oil palm
and almost daily for rubber; thus, soils along the harvest path and part of the death path are
very compacted. The soils under oil palm and rubber plantations remain unploughed for the
entire growing period. Weeds in the oil palm plantations are regularly eradicated using
herbicides or mechanical equipment. Intensive inorganic fertilization ($1000 \text{ kg ha}^{-1} \text{ year}^{-1}$)
also occurs, contributing to the degradation of the soil structure and fauna.
**2.2 SWAT model**
The SWAT model is a continuous long-term yield model that was developed to simulate the
impact of different land cover/management practices on streamflow in complex watersheds
with varying soil, land use, and management conditions over long time periods. The major
model components include weather, hydrology, soil temperature, soil properties, plant
growth, nutrients, and land management (Neitsch et al., 2011; Arnold et al., 2012).
During the modeling process, a watershed is subdivided into several sub-watersheds,
which are then further partitioned into hydrological response units (HRUs) that are defined
by their topography, soil, and land use characteristics, which are not spatially referenced in
the model. The hydrological outputs of HRUs are calculated using the water balance equation
and include total streamflow, surface flow and baseflow. These output components can then
be used to calculate indicators of the water flow regulation functions of a watershed, namely
C, which is the ratio of direct runoff to rainfall, and BFI, which is the proportion of baseflow
in the streamflow.
Because the SWAT model was designed for temperate regions, adapting crop parameter
inputs for use in tropical regions is necessary (Strauch et al., 2013; Van Griensven et al.,
2014, Alemayehu et al., 2017). To avoid incorrect parameterization of sensitive values, we
carried out field measurements for interception, infiltration, and surface runoff to adapt the
parameter values, particularly those that control surface runoff and baseflow processes. We
then performed SWAT model simulation in two study watersheds and conducted small
watershed experiments to compare the observed C values with those obtained from
simulations.
**2.2.1 Model setup**
Delineation of watersheds and their sub-watersheds in our study area was automatically
performed by the SWAT model and was based on a digital elevation model (DEM) with a 30
m resolution. During this automatic delineation, we pre-defined 50 000 ha as a threshold for
the minimum sub-watershed area, based on subdivision of the BH and MT watersheds into 25
and 23 sub-watersheds, respectively.
**Crop parameters**
Oil palm plantations exhibit specific characteristics, particularly with respect to rainfall
partitioning. These characteristics include high interception, high ET, low soil infiltration,
high proportion of surface runoff, and absence of leaf litter, of which the first four can
potentially reduce baseflow. Therefore, to consider these specific characteristics, we
conducted field measurements and adjusted several crop parameters that are related to flow
components, including canopy storage (CANMX), plant uptake compensation factor (EPCO),
hydrologic soil group (HSG) and Soil Conservation Service (SCS) curve number (CN).
**a) Interception**
CANMX is the maximum amount of water that can be stored in the canopy and trunks of
fully developed trees. Thus, an increase in this parameter reflects a reduction in the amount of
rainfall that reaches the ground. In oil palms, rainfall is intercepted not only by leaves and
branches but also by water reservoirs in leaf axils along the trunk. Therefore, we measured
the water storage capacity of leaf axils along the trunks of four 10–12 years old oil palm trees
with 10 replications per tree. We found that the leaf axils along the trunk can store up to 20 L
or 8.4 mm of water (Table 1), which matches previous reports that the leaf axils along oil
palm trunks have a high water storage capacity (Merten et al., 2016; Meijide et al., 2017).
Table 1. Water storage capacity of the leaf axils along the trunks of oil palm trees

| Replicate | Water storage (mm) | | | |
|---|---|---|---|---|
| | Tree 1 | Tree 2 | Tree 3 | Tree 4 |
| 1 | 14.2 | 10.8 | 4.4 | 6.2 |
| 2 | 10.6 | 10.2 | 4.2 | 5.9 |
| 3 | 9.4 | 10.5 | 5.9 | 7.9 |
| 4 | 8.1 | 10.4 | 5.4 | 7.4 |
| 5 | 8.8 | 11.5 | 3.8 | 9.8 |
| 6 | 9.4 | 10.9 | 6.0 | 8.0 |
| 7 | 9.3 | 10.6 | 5.9 | 7.5 |
| 8 | 10.1 | 11.0 | 5.2 | 7.3 |
| 9 | 8.9 | 11.2 | 4.7 | 10.5 |
| 10 | 9.5 | 11.3 | 4.9 | 7.2 |
| Average = 8.4 mm | | | | |

We also measured the canopy interception by oil palm, rubber, agroforest, and forest
canopies between November 2012 and February 2013. Rainfall interception was assessed by
measuring throughfall, stemflow, and subtracting these from the incident rainfall. In total,
there were 30 rainfall events during this time, representing light-to-heavy rain. We found that
oil palm plantations tended to exhibit higher levels of canopy interception (Fig. 2), with our
estimates falling within the range of values that were previously reported for tropical forests
in Southeast Asia (commonly 10 %–30 %; Kumagai et al., 2005; Dietz et al., 2006). These
interception assessments were used to adjust the CANMX parameter of the SWAT model
(Table 2).

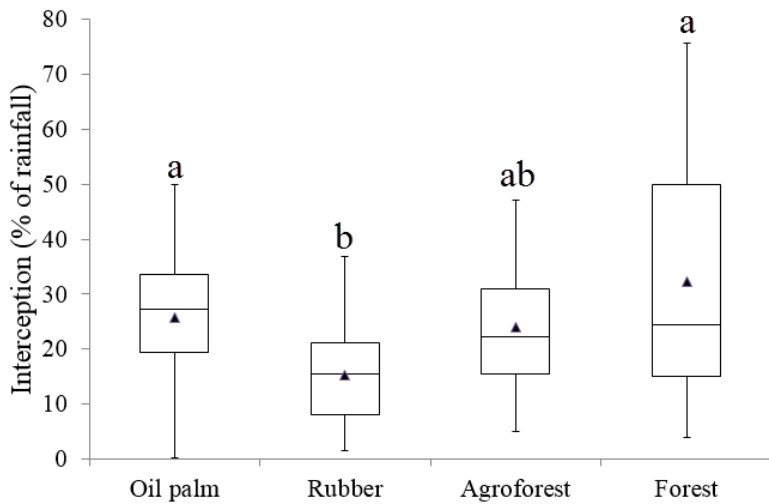


Figure 2. Canopy interception of rainfall under different land uses. Different letters indicate
significant differences between the means (Bonferroni-corrected post hoc t-test based on
ANOVA; $p < 0.05$).
**b) ET**
Actual ET was determined by measuring the daily depletion of soil moisture content at a
distance of 2 m from the trunks of oil palm trees. Soil moisture measurements were made on
consecutive no rain days over the 16-day period from 25 July, 2012 to 10 August, 2012. On
an average, soil moisture decreased by 6 % (vol) over this period, which is equivalent to 72
mm or 4.5 mm day$^{-1}$ and is relatively high compared with the average land use in the study
area. Similarly, Meijide et al. (2017) also reported a yearly oil palm ET of 1 216 mm (4.7 mm
day$^{-1}$) using Edy covariance measurements in the study area. This ET rate is similar to or
even higher than ET rates for forests in Southeast Asia (Kumagai et al., 2005), despite oil
palm having a much lower stand density and biomass per hectare. Therefore, we accordingly
adjusted EPCO, which is related to ET (Table 2).
**c) Infiltration and surface runoff**
One important parameter of the SWAT model that is related to surface runoff is CN (Arnold
et al., 2012), which determines the proportion of rainfall that becomes surface runoff (range,
0–100, with a higher value reflecting a higher level of surface runoff). The CN value is
grouped into four HSGs (i.e., A, B, C, and D) according to the soil infiltration capacity. To
adjust the CN value, we measured soil infiltration and surface runoff in each land use type
(i.e., oil palm, rubber, agroforest, and forest) using a double-ring infiltrometer and
multidivisor runoff collectors mounted at the lower end of each plot, respectively. The
infiltration rate in different land-use types increases in the following order: oil palm harvest
path (3 cm h$^{-1}$) < oil palm circle (3 cm h$^{-1}$) < rubber harvest path (7 cm h$^{-1}$) < between rubber
trees (7.8 cm h$^{-1}$) < forest (47 cm h$^{-1}$). The infiltration in the oil palm, rubber plantations were
markedly lower than those at the forest. Low infiltration rate in the oil palm is associated with
the soil compaction due to the intensive harvest activities.
For all HRUs with oil palm and rubber land uses, we selected HSG category D (Table 2)
owing to its high surface runoff and low infiltration rate. We assumed that CN values of the
forest and agroforest were similar to those of the evergreen and mixed forest, respectively, in
the SWAT crop database.
**d) Litter fall**
In the oil palm plantations, negligible litter was found outside the frond piles. Litter fall in oil
palm plantations does not naturally occur, but leaves are cut during fruit harvest and piled up
in a frond pile, which occupies only 12 % of the entire oil palm plantation area.
Consequently, the ground surface of oil palm plantations is managed mostly without litter,
leading to higher surface runoffs. There was also negligible understory vegetation (grasses)
because herbicides were routinely sprayed. The absence of the litter fall affected Manning's
"n" value for overland flow (OV_N).
Table 2. Adapted parameter inputs for the study area

| SWAT parameter | Definition | Oil palm | Rubber | Agroforest | Forest |
| --- | --- | --- | --- | --- | --- |
| CANMX (mm) | Maximum trunk storage | 8.4 | 0 | 0 | 0 |
| | Maximum canopy storage | 4.7 | 2.7 | 4.3 | 5.8 |
| HYDGRP | Hydrologic soil group | D | D | B | C |
| CN2 | Curve number | 83 | 83 | 65 | 45 |
| SOL_BD (g cm$^{-3}$) | Soil bulk density | 1.2–1.3 | 1.2–1.3 | 1 | 0.9 |
| EPCO | Plant uptake compensation factor | 1 | 1 | 1 | 1 |
| OV_N | Manning's "n" value for overland flow | 0.07 | 0.14 | 0.4 | 0.5 |
| SOL_K (mm h$^{-1}$) | Saturated hydraulic | 30 | 78 | 400 | 470 |

| | | | | | |
|---|---|---|---|---|---|
| | conductivity | | | | |
| SOL_AWC (mm mm$^{-1}$) | Available water capacity | 0.1 | 0.1 | 0.2 | 0.2 |
| BLAI (m$^2$ m$^{-2}$) | Maximum potential leaf area index | 3.6[a] | 2.6 | 5 | 5 |
| CHTMX (m) | Maximum canopy height | 12 | 13 | 14 | 20 |
| T_BASE | Base temperature | 20 | 20 | 20 | 20 |
| ALPHA_BF | Baseflow recession constant | 0.90-0.95 | 0.90-0.95 | 0.90-0.95 | 0.90-0.95 |
| T_OPT | Optimal temperature | 28 | 28 | 30 | 30 |

[a]Fan et al., 2015
**e) Baseflow**
The baseflow recession constant (ALPHA_BF) was calculated by plotting the selected daily
streamflow hydrograph on semi-log paper and determining the average values from several
individual rainfall events. Previous study (Tarigan et al., 2016b), showed similar range of
ALPHA-BF values in the study area.
**General input data**
The SWAT model requires considerable other types of input data in addition to the crop
parameters described above, such as the climate, topography, soil type, and land use for each
sub-watershed (Table 3).
Table 3. Model input data sources

| Data type | Resolution | Description | Source |
|---|---|---|---|
| Topography | 30 m | DEM with a resolution of 30 m per pixel | SRTM |

| | | | |
|---|---|---|---|
| Soil map | 1:250 000 | Additional soil data were collected from the field and previous studies | Soil Research Institute, Ministry of Agriculture |
| Land use | 1:100 000 | Land use map with intensive ground check | Regional Planning office (BAPPEDA[a]) |
| Rainfall and climate | Daily | Rainfall and meteorological stations at Rantau Pandan, Siulak Deras, Muara Hemat, Padang Aro, Depati Parbo, Bangko Bungo, Pematang Kabau, and Bungku | BMKG[b] office and CRC990[c] |
| Streamflow | Daily discharge data | Stations at Muara Tembesi, Air Gemuruh, Batang Tabir, Batang Pelepat, and Muara Kilis | Ministry of Public Works (BBWS[d]) |


[a]BAPPEDA, Regional Planning Agency (*Badan Perencanaan Daerah*); [b]BMKG,
Meteorology, Climatology and Geophysics Agency (*Badan Meteorologi, Klimatologi dan*
*Geofisika*); [c]CRC990, Collaborative Research Centre 990; [d]BBWS, Ministry of Public Works
(*Balai Besar Wilayah Sungai*).
DEM with a resolution of 30 m per pixel was derived from NASA Shuttle Radar
Topography Mission (SRTM). Soil map was obtained from Soil Research Institute at a scale
of 1: 250 000. Some soil parameters such as soil hydraulic conductivity, bulk density,
available water content, and texture were derived from previous study (Sunarti et al., 2008).
Additional soil data were collected from Batang Tabir sub-watershed and from CRC990 plots
in Bukit Duabelas and Hutan Harapan landscape (Drescher et al., 2016). Daily rainfall and
climate data between 2000 and 2014 were sourced from the rainfall and meteorological
stations at Rantau Pandan, Siulak Deras, Muara Hemat, Padang Aro, Depati Parbo, Bangko
Bungo, Pematang Kabau, and Bungku. Daily streamflow data between 2000 and 2014 were
provided by the Ministry of Public Works (BBWS). All these data are freely available for
research purposes on official request to the corresponding institutions. The streamflow time
series and rainfall records for the small catchments and the soil data have been deposited by
the first author at Bogor Agricultural University and in the EFForTS Database (https://efforts-
is.uni-goettingen.de). The land use for the study area was obtained from Jambi Province
Regional Planning and Agricultural Plantation offices (Ditjenbun, 2015).
**2.2.2 Model validation and calibration**
The first step in the calibration and validation process in SWAT is the determination of the
most sensitive parameters for a given watershed (Van Griensven et al., 2006; Arnold et al.,
2012). The sensitivity analysis was performed using the SWAT Calibration and Uncertainty
Procedure (SWAT-CUP) package, which is an interface for auto-calibration that was
specifically developed for SWAT and which links any calibration/uncertainty or sensitivity
program to SWAT (Abbaspour, 2015).
Following the sensitivity analysis, we calibrated the SWAT model using the Latin
hypercube sampling approach of the SWAT-CUP software. We first determined parameter
ranges based on the minimum and maximum values allowed in SWAT. We then performed
calibration and validation of the SWAT model by comparing the simulated monthly
streamflows with observed data at the Muara Kilis and Muara Tembesi gauging stations from
2007 to 2009 for calibration and 2012 to 2014 for validation. Moriasi et al. (2007; 2015)
recommended the use of three quantitative statistics for model evaluation: the Nash–Sutcliffe
efficiency (NSE), percent bias (PBIAS), and ratio of the root mean square error to the
standard deviation of measured data. In this study, we used NSE and PBIAS to evaluate the
model performance which is consistent with the majority of the existing SWAT literature
(Gassman et al., 2007; Douglas-Mankin et al., 2010; Tuppad et al., 2011; Gassman et al.,
2014; Bressiani et al., 2015). NSE is a normalized statistic that determines the relative
magnitude of the residual variance ("noise") compared with the measured data variance
("information") (Nash and Sutcliffe, 1970). PBIAS measures the average tendency of
simulated data to be larger or smaller than that of observational data (Gupta et al., 1999), with
an optimum value of zero and lower values that indicate better simulations. Positive values of
PBIAS indicate model underestimation, whereas negative values indicate model
overestimation.
**2.3 Simulated C and BFI values and the proportion of land use types in a watershed**
The output of the validated SWAT model consisting of flow components for each sub-
watershed was used to calculate indicators of the water flow regulation functions of a
watershed, namely C, which is the ratio of direct runoff to rainfall, and BFI, which is the
proportion of baseflow in the streamflow. To analyze the association between the C and BFI
values and the proportion of each land-use type in a watershed, we derived data vectors from
the BH and MT watersheds. Each of these vectors corresponded to the percentage of land use
types, C and BFI in each of the 25 sub-watersheds from the BH watershed and 23 sub-
watersheds from the MT watershed (Fig. 3a and b; Table 4).

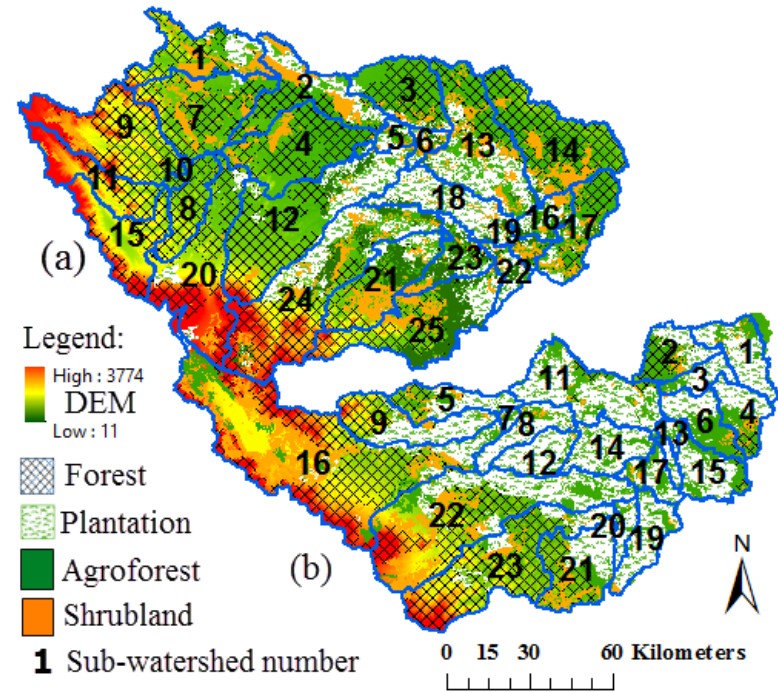


Figure 3. Land use types and sub-watershed number of the (a) BH and (b) MT watersheds.
Table 4. The percentage of land use types, C and BFI in each sub-watershed within the BH
and MT watersheds

| Sub-wat. nr. | Percentage of land use types | | | | | C | BFI |
|---|---|---|---|---|---|---|---|
| | F | AF | RP | OP | S | | |
| BH watershed | | | | | | | |
| 1 | 56 | 0 | 17 | 0 | 26 | 0.21 | 0.71 |
| 2 | 46 | 0 | 26 | 0 | 26 | 0.27 | 0.63 |
| 3 | 90 | 0 | 0 | 0 | 0 | 0.00 | 0.99 |
| 4 | 76 | 0 | 0 | 0 | 0 | 0.18 | 0.75 |
| 5 | 16 | 0 | 67 | 0 | 17 | 0.40 | 0.32 |
| 6 | 0 | 0 | 67 | 0 | 34 | 0.41 | 0.23 |

| Sub-wat. nr. | Percentage of land use types | | | | | C | BFI |
|---|---|---|---|---|---|---|---|
| | F | AF | RP | OP | S | | |
| 24 | 66 | 5 | 25 | 5 | 0 | 0.14 | 0.76 |
| 25 | 44 | 32 | 0 | 7 | 17 | 0.04 | 0.93 |
| MT watershed | | | | | | | |
| 1 | 0 | 13 | 0 | 85 | 0 | 0.59 | 0.11 |
| 2 | 56 | 0 | 0 | 44 | 0 | 0.36 | 0.45 |
| 3 | 0 | 16 | 0 | 82 | 0 | 0.58 | 0.13 |
| 4 | 21 | 20 | 12 | 48 | 0 | 0.45 | 0.32 |

| 7 | 68 | 0 | 0 | 0 | 19 | 0.17 | 0.76 | 5 | 32 | 0 | 19 | 38 | 12 | 0.40 | 0.29 |
| 8 | 100 | 0 | 0 | 0 | 0 | 0.01 | 0.98 | 6 | 0 | 46 | 0 | 52 | 0 | 0.43 | 0.37 |
| 9 | 80 | 0 | 0 | 0 | 0 | 0.15 | 0.79 | 8 | 0 | 0 | 15 | 86 | 0 | 0.54 | 0.01 |
| 11 | 55 | 0 | 0 | 0 | 0 | 0.31 | 0.57 | 9 | 48 | 0 | 0 | 53 | 0 | 0.59 | 0.11 |
| 12 | 61 | 0 | 23 | 0 | 0 | 0.23 | 0.62 | 10 | 0 | 0 | 0 | 96 | 0 | 0.34 | 0.37 |
| 13 | 30 | 0 | 41 | 0 | 18 | 0.32 | 0.42 | 11 | 0 | 0 | 19 | 69 | 0 | 0.67 | 0.00 |
| 14 | 84 | 0 | 0 | 0 | 16 | 0.04 | 0.91 | 12 | 0 | 0 | 13 | 88 | 0 | 0.66 | 0.01 |
| 15 | 49 | 0 | 0 | 0 | 0 | 0.31 | 0.58 | 14 | 0 | 0 | 20 | 80 | 0 | 0.54 | 0.01 |
| 16 | 0 | 20 | 39 | 0 | 24 | 0.36 | 0.35 | 15 | 0 | 27 | 14 | 57 | 0 | 0.65 | 0.01 |
| 17 | 59 | 0 | 0 | 0 | 24 | 0.16 | 0.71 | 16 | 70 | 0 | 0 | 0 | 31 | 0.53 | 0.22 |
| 18 | 0 | 0 | 82 | 19 | 0 | 0.48 | 0.03 | 17 | 0 | 28 | 11 | 57 | 0 | 0.28 | 0.65 |
| 19 | 19 | 30 | 0 | 0 | 52 | 0.13 | 0.75 | 19 | 0 | 12 | 31 | 57 | 0 | 0.53 | 0.23 |
| 20 | 69 | 0 | 0 | 0 | 0 | 0.22 | 0.69 | 20 | 0 | 0 | 63 | 37 | 0 | 0.60 | 0.10 |
| 21 | 45 | 24 | 0 | 0 | 32 | 0.08 | 0.87 | 21 | 48 | 0 | 0 | 36 | 16 | 0.65 | 0.01 |
| 22 | 0 | 26 | 16 | 58 | 0 | 0.35 | 0.28 | 22 | 36 | 0 | 17 | 17 | 16 | 0.32 | 0.42 |
| 23 | 0 | 48 | 33 | 0 | 20 | 0.23 | 0.61 | 23 | 100 | 0 | 0 | 0 | 0 | 0.40 | 0.32 |

Sub-wat. nr, sub-watershed number (see Fig. 3a and b); F, forest; AF, agroforest; RP, rubber
plantation; OP, oil palm; S, shrubland.
**2.4 Measured C values**
We carried out the direct C measurements from the two small watersheds (Fig. 1c) between
2013 and 2015 using rectangular weirs and water level recorders for comparison with
simulated C values. The land use in the small watersheds is similar to the proportions found
in several of the sub-watersheds shown in Table 4 (e.g., BH 18, MT 1, MT 8, MT 10, MT 11,
MT 12, MT 14, and MT 20). The direct runoff components of the hydrographs were
separated using the straight-line method described by Blume et al. (2007), following which C
was calculated. We did not calculate BFI values along with C values in the small watershed
experiments because BFI calculation requires hydrograph records over a longer period.
**3  Results and Discussion**

## 3.1 Measured C values

The average C value was based on nine individual rainfall events during the field experiment. The observed C values that were measured during two small watershed experiments was 0.59 (Table 5). This value was comparable with the averaged simulated values of 0.60 for sub-watersheds with comparable proportions of land use type with those of the small watershed experiment (Table 6).

Table 5. Observed C values obtained from field experiments in two small watersheds that were dominated by plantation cover

| Event Nr. | Rainfall (cm h$^{-1}$) | Rainfall volume (m$^3$) | | Runoff (m$^3$) | | C | |
|---|---|---|---|---|---|---|---|
| | | Small wat. 1 | Small wat. 2 | Small wat. 1 | Small wat. 2 | Small wat.1 | Small wat. 2 |
| 1 | 6.0 | 8960 | 3136 | 4500 | 1320 | 0.50 | 0.42 |
| 2 | 3.0 | 5180 | 1813 | 2625 | 840 | 0.51 | 0.46 |
| 3 | 1.4 | 4095 | 1433 | 3000 | 810 | 0.73 | 0.57 |
| 4 | 0.6 | 1456 | 509.6 | 1080 | 255 | 0.74 | 0.50 |
| 5 | 10.7 | 14923 | 5223 | 11250 | 3900 | 0.75 | 0.75 |
| 6 | 3.1 | 6006 | 2102 | 4050 | 1020 | 0.67 | 0.49 |
| 7 | 2.3 | 8188 | 2885 | 6150 | 1584 | 0.75 | 0.55 |
| 8 | 2.2 | 4416 | 1465 | 2400 | 780 | 0.54 | 0.53 |
| 9 | 9.6 | 8916 | 3121 | 5400 | 1650 | 0.61 | 0.53 |
| Average | | | | | | 0.65 | 0.53 |
| Total | | | | | | 0.59 | |

Table 6. Simulated C values for the sub-watersheds used in the SWAT simulation (Table 4) that had similar percentages of plantation cover to the two small watersheds used in the field experiments

| Sub-watershed code (see Table 4)[a] | Percentage of plantation cover | | C |
|---|---|---|---|
| | Oil palm | Rubber | |
| BH-18 | 82 | 19 | 0.49 |
| MT-1 | 85 | 0 | 0.59 |
| MT-8 | 86 | 15 | 0.54 |

| | | | | |
|---|---|---|---|---|
| MT-10 | 96 | 0 | 0.67 | |
| MT-11 | 69 | 19 | 0.67 | |
| MT-12 | 88 | 13 | 0.54 | |
| MT-14 | 80 | 20 | 0.65 | |
| MT-20 | 37 | 63 | 0.65 | |
| Average | | | 0.60 | |

[a]Sub-watersheds were selected that had >80 % plantation (oil palm and rubber) cover,
allowing a comparison to be made with the small watersheds used in the field experiment
**3.2 SWAT model performance**
The sensitive parameters that were included in the calibration of the SWAT model are ranked
in Table 7. Some of these parameters play an important role in controlling the initial
abstraction of rainfall (e.g., CANMX), rainfall partitioning into surface runoff (e.g., CN2 and
OV_N), and vertical movement of water through the soil (e.g., SOL_BD, SOL_K, and
SOL_AWC).
Table 7. Sensitivity rank, initial and final values of the calibration parameters that were used
in the study for the BH and MT watersheds

| Parameter | Description | Sensitivity rank | Initial value range | Best-fit values | |
|---|---|---|---|---|---|
| | | | BH and MT | BH | MT |
| ALPHA_BF | Baseflow recession constant | 1 | 0.0 to 1.0 | 0.94 | 0.91 |
| CN2 | SCS runoff curve number for moisture condition II | 2 | −0.2 to 0.2 (V)[a] | 0.14 | 0.12 |
| GW_DELAY | Groundwater delay time (days) | 3 | 30 to 450 | 62.5 | 57.2 |
| CANMX | Maximum canopy storage (mm) | 4 | −0.2 to 1.0 (V)[a] | 0.95 | 0.76 |
| SOL_BD | Soil bulk density | 5 | −0.5 to 0.6 (V)[a] | 0.46 | 0.47 |
| GWQMN | Water depth in a shallow aquifer for a return flow (mm $H_2O$) | 6 | 0.0 to 2.0 | 0.99 | 0.95 |
| SOL_K | Saturated hydraulic conductivity | 7 | −0.8 to 0.8 (V)[a] | 0.71 | 0.62 |

| | | | | | |
|---|---|---|---|---|---|
| | (mm h$^{-1}$) | | | | |
| CH_N2 | Manning's "n" value for the main channel | 8 | 0.0 to 0.3 | 0.05 | 0.15 |
| SOL_AWC | Available water capacity of the soil (mm H$_2$O/mm soil) | 9 | −0.2 to 0.4 (V)[a] | 0.09 | 0.04 |
| OV_N | Manning's "n" value for overland flow | 10 | −0.2 to 1.0 (V)[a] | 0.51 | 0.3 |

[a] (V) = Variable depends on land use and soil, and so changes in calibration were expressed
as a fraction.
A visual comparison of best-fit simulations and observed data is shown in Fig. 4, with
NSE values of 0.80–0.88 (calibration) and 0.84–0.85 (validation) and PBIAS values of −2.9
to 1.2 (calibration) and 7.0–11.9 (validation) for BH and MT watersheds, respectively. Based
on the criterion proposed by Moriasi et al. (2007; 2015), the model performance was
considered very good and satisfactory for calibration and validation respectively.

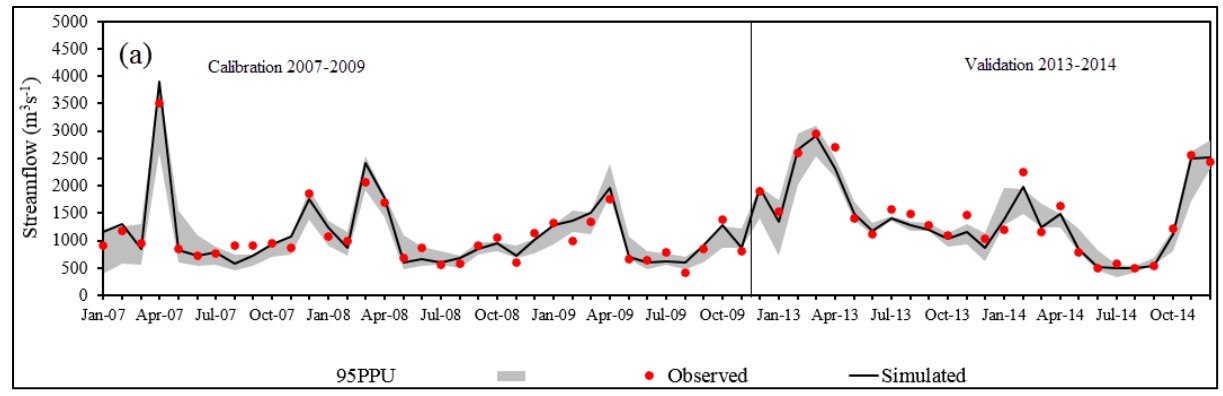


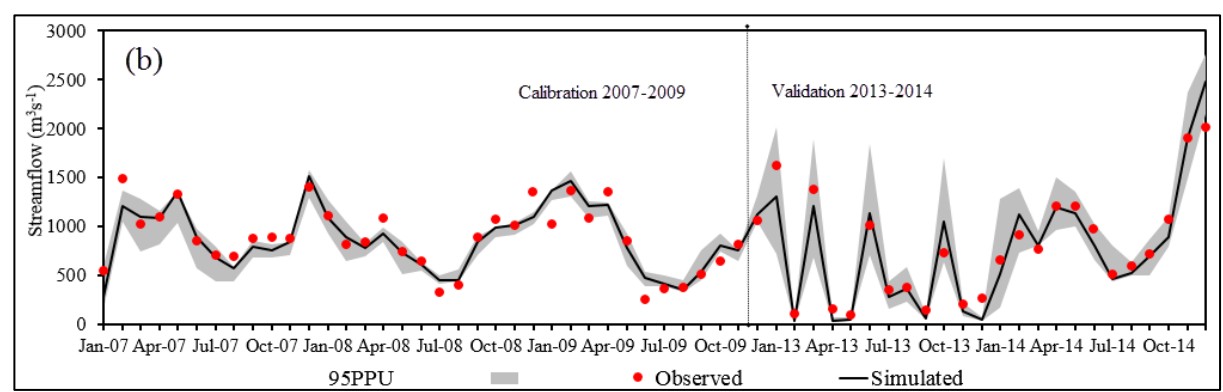

Figure 4. Observed vs. simulated streamflow and 95 % uncertainty interval (95PPU; see

Abbaspour, 2015) with P factors of 0.83 and 0.76 and R factors of 0.65 and 0.69 for the (a)

BH and (b) MT watersheds, respectively.

**3.3 Simulated C and BFI values and the proportion of land use types in a watershed**

The proportion of a particular land use type in a sub-watershed was significantly correlated

with C and BFI values obtained from 48 data vectors (Table 4). C values significantly

decreased as the percentage of forest cover increased ($R^2 = 0.73$; $p < 0.05$; Fig. 5a) and

significantly increased as the percentage of plantation cover increased ($R^2 = 0.74$; $p < 0.05$;

Fig. 5b). Low infiltration capacity in oil palm and rubber plantations was the reason for

higher C values in the sub-watersheds with high proportions of the plantation land use. There

were no significant associations between C values and any of the other land use types such as

shrubland (Fig. 5c), agroforest (Fig. 5d), and dryland farming (data not shown). Some sub-

watersheds had low C values despite having low levels of forest cover (e.g., BH 19). This can

be explained by the fact that BH 19 had no oil palm or rubber plantations but consisted of 52
% of shrubland, which will have helped in reducing the C value. Furthermore, some
watersheds had 100 % forest cover but a low BFI value (e.g., MT 23). This is the only sub-
watershed in the MT watershed to have a high proportion of steeper slopes (76 % of the sub-
watershed), which will increase C and decrease BFI. Among the 48 sub-watersheds we
considered, only two had these slope characteristics.

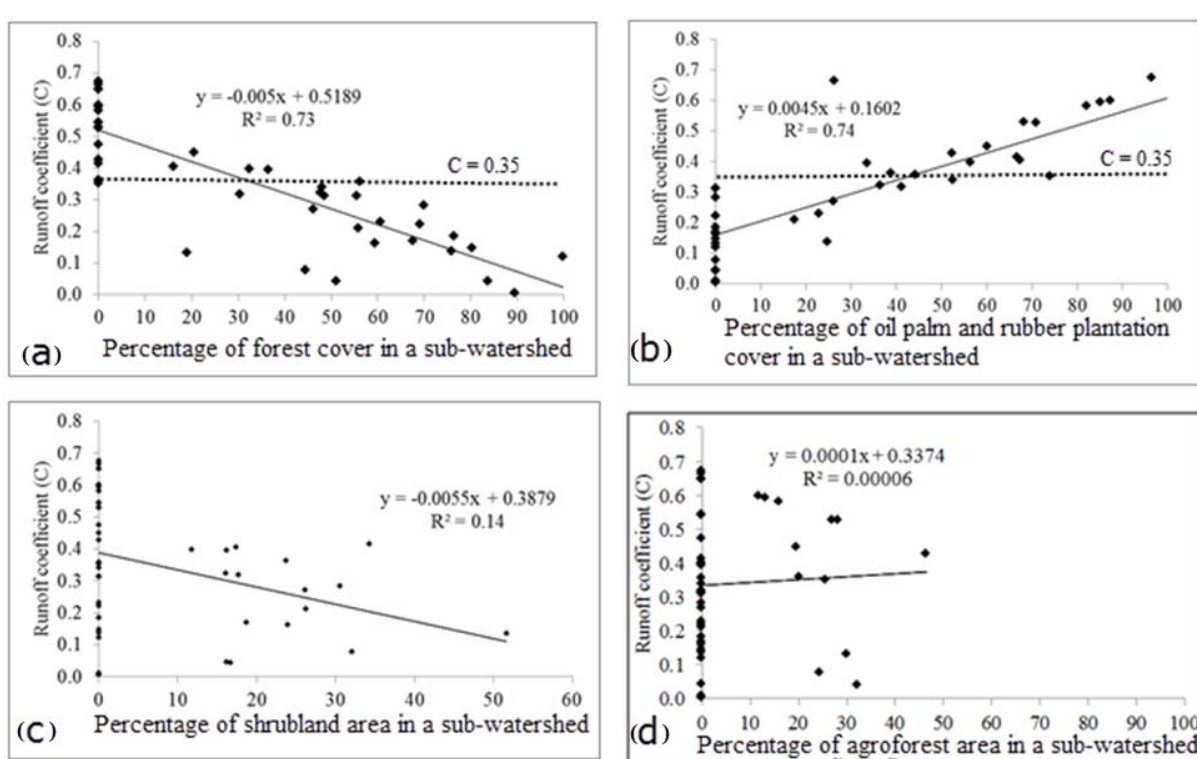


Figure 5. Association between simulated C values and the percentage of each land use type in
a particular sub-watershed. Dotted lines indicate the maximum acceptable C value according
to the Ministry of Forestry Decree (2013).
The Ministry of Forestry of Indonesia considers C values of $< 0.35$ to be adequate for
supporting the required ecosystem services of Indonesian watersheds (Ministry of Forestry
Decree, 2013). Based on our findings, $\geq 30$ % forest cover (Fig. 5a) and $\leq 40$ % plantation
cover (Fig. 5b) are required in a given sub-watershed with rapid expansion of plantation to
achieve the desired C value.

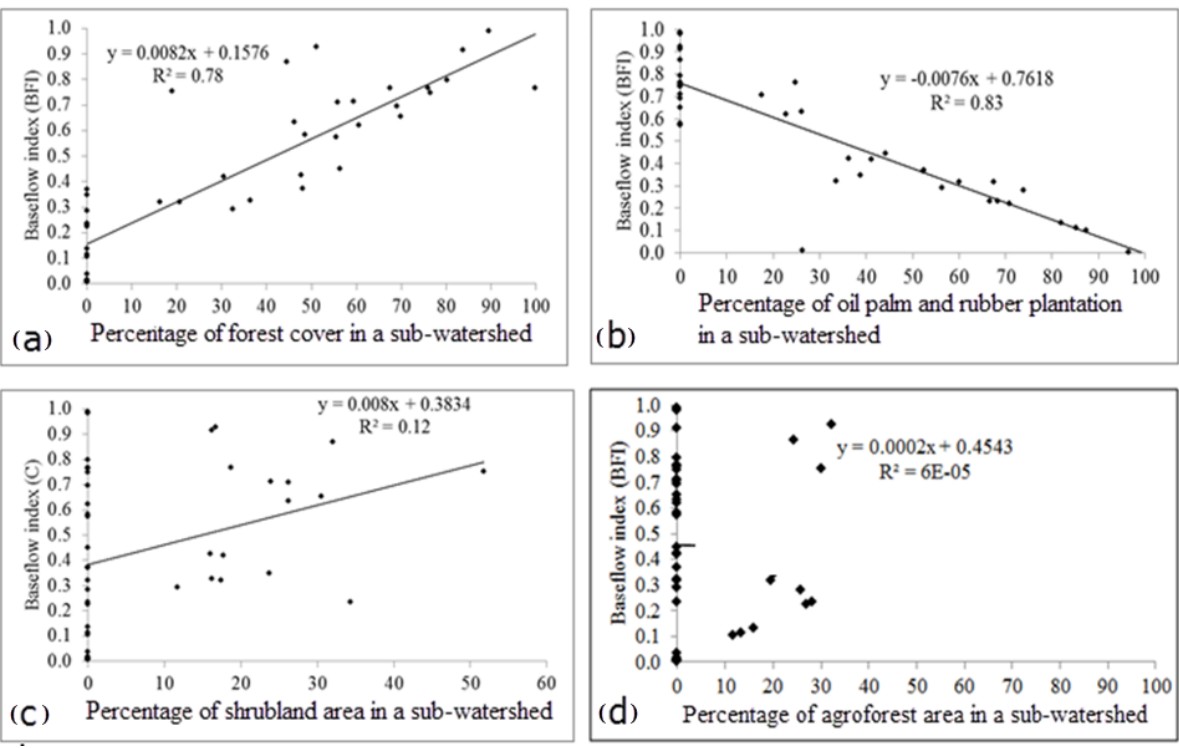


Figure 6. Association between simulated BFI values and the percentage of each land use type
in a given sub-watershed.

BFI values significantly increased as the percentage of forest cover increased ($R^2 = 0.78$; $p$

$< 0.05$; Fig. 6a) and significantly decreased as the percentage of plantation cover increased
($R^2 = 0.83$; $p < 0.05$; Fig. 6b). BFI was not significantly related to any other land use types
such as shrubland (Fig. 6c), agroforest (Fig. 6d), and dryland farming (data not shown).

According to Neitsch et al. (2009), the SWAT model considers only shallow groundwater

in stream flow simulation. Therefore, we expected that SWAT underestimated BFI values in
our study area. To improve the performance of the SWAT model for deep groundwater flow
(low flow) simulation, Pfannerstill et al. (2014) modified groundwater module by splitting the
active groundwater storage into a fast and a slow contributing aquifer.  Similar studies that
focused on modifications of the SWAT groundwater component to obtain improved baseflow
and overall streamflow results have also been  reported by Luo et al. (2012) and Wang and
Brubaker (2015). Similar modifications are needed in the standard SWAT model in order to
more accurately simulate the conditions such as those encountered in this study.
**3.4 Application of the research findings**

The conversion of tropical rainforest into oil palm and rubber plantations affects the local

hydrological cycle by increasing ET, decreasing infiltration, decreasing low flow levels, and
increasing flooding frequency. In Jambi Province, Indonesia, forested areas have been largely
transformed into plantations, resulting in inhabitants experiencing water shortages during the
dry season and dramatic increases in flooding frequency during the wet season. One way in
which this problem could be mitigated is by maintaining an adequate proportion of forested
and plantation areas in a particular watershed, but this raises the question about what is the
minimum percentage of forest area and the maximum proportion of plantation area in a
watershed that will allow the maintenance of adequate water flow regulation. This study is
the first to describe the quantitative association between forest and plantation areas and the
flow indicators C and BFI; this understanding is required by spatial planners if they are to
balance the ecology and socioeconomic functions of a landscape with the rapid expansion of
plantation crops. In addition, our study provides data regarding how SWAT input parameters
related to tropical plantations such as oil palm and rubber should be adjusted, particularly
those that play an important role in controlling rainfall initial abstraction (e.g., CANMX),
rainfall partitioning into surface runoffs (e.g., CN2, OV_N), and vertical movement of water
through (e.g., SOL_BD, SOL_K, and SOL_AWC).
**4 Summary**
We found that ALPHA_BF, CN2, GW_DELAY, CANMX, SOL_BD, GWQMN, SOL_K,
CH_N2, SOL_AWC, and OV_N were sensitive parameters in our model, some of which play
an important role in controlling the initial abstraction of rainfall (e.g., CANMX), rainfall
partitioning into surface runoff (e.g., CN2, OV_N), and vertical movement of water through
the soil (e.g., SOL_BD, SOL_K, and SOL_AWC).
Overall, the SWAT model performance was strong, with NSE values of 0.80–0.88
(calibration) and 0.80–0.85 (validation) and PBIAS values of −2.9 to 1.2 (calibration) and
7.0–11.9 (validation). We found that the percentage of forest cover in a watershed was
significantly negatively correlated with C and positively correlated with BFI, whereas the
percentage of rubber and oil palm plantation cover showed the opposite pattern. Finally, our
findings suggest that a watershed should contain ≥ 30 % forest cover and a maximum of 40
% plantation cover for maintaining sustainable water flow regulation ecosystem services.
The quantitative association between forest cover and flow indicators, which was derived
in this study, will help regional planners in determining the minimum proportion of forest
cover that needs to be maintained to ensure effective water flow regulation in a watershed.
**5. Data availability**
The land use data are freely available for research purposes upon official request to the
corresponding institutions: the rainfall and climate data can be obtained from the
Meteorology and Geophysics Agency; the streamflow data of the macro watersheds can be
obtained from the Ministry of Public Works; and the land use data can be obtained from the
Regional Planning office. The streamflow time series and rainfall records for the small
watersheds and data for the resampled soil hydraulic conductivity, bulk density, available
water content and texture have been deposited by the first author at Bogor Agricultural
University and in the EFForTS Database (https://efforts-is.uni-goettingen.de).

**Acknowledgments**

This study was performed in the framework of the joint Indonesian-German research project
EFForTS-CRC 990 (http://www.uni-goettingen.de/crc990) and was funded by the Directorate
General of Higher Education (DIKTI), Indonesia.

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
