# Peer review of "Minimum forest cover required for sustainable water flow regulation of a watershed: a"

_Hydrology and Earth System Sciences, 2017_

## Short Comment (SC1) · 13 Jun 2017

Dear Mr/Mrs/Ms

This research is very important for the forestry sector, especially to the government of Indonesia in making policy, among others: 1. Has been in line with the Indonesian government's policy to establish forest cover area in a watershed minimum 30%. 2. For monoculture plants has not been applied policies on the right area because the mono-culture plant is very large influence on the flow of the surface and its impact on erosion and sedimentation. 3. Indonesia is currently faced with monoculture, especially palm and rubber 4. Hopefully this research will have a wider impact not only in Indonesia but

also in other countries, especially in tropical countries.

Best regards,

Onesimus Patiung Indonesia

————————————————

---

## Short Comment (SC2) · 14 Jun 2017

1. This topic is very relevant because the research on the problem under study is still rare in Indonesia, but it has lots of benefits for the survival of living things in the earth. The results of this study are very useful for planners to establish forest areas that should be released to be used as plantation areas such as oil palm or rubber plantations. 2. The water flow regulation in the watershed is influenced by several factors such as land cover, number and distribution of rainfall, soil type, topography, and the geology of the place concerned. 3. Although Indonesia declared as an area with tropical rain forest but in the reality rainfall pattern in each location so varied therefor the

research like this still needed in some different place. 4. SWAT models need to be tested in other areas and the results should be compared with the reality of water availability in the area concerned. 5. Evapotranspiration measurements carried out using lysimeters in several places in Indonesia show that similar species evapotranspiration in several locations are not exactly the same.

———————————————

---

## Short Comment (SC3) · 1 Jul 2017

The study presented in the paper analyzes minimum forest cover in a watershed in order to maintain sustainable water flow regulation under a rapid expansion of plantation developments. This is an important paper as such a study is still lacking in Indonesia where the expansion of oil palm plantations has occurred rapidly in the past few years converting most of previously forest lands and reducing the total area of forest cover. This condition will potentially decrease the watershed's health and its function to regulate water flow if the land cover or land use type is not properly regulated.

This study will expectedly fill a gap in our understanding on how forests have con-

tributed to the water flow regulation along with some other factors such as rainfall, soil type, land management practices, and other influencing factors in the watershed. My suggestion is that to test the SWAT hydrological model that has been employed in this study in other areas with varying watershed characteristics and conditions to evaluate the indicators of water flow regulation function with various land covers or land-use types.

The method and result of this study will be useful input for stakeholders involved in spatial planning to define the minimum proportion of forest cover area in the watershed to maintain sustainable water flow regulation and what maximum area could be allocated for other land use types.

---

## Short Comment (SC4) · 2 Jul 2017

The manuscript addresses one of the important ecosystem services (i.e. water regulation function) that was observed and modeled in a land use system mostly dominated by oil palm and rubber plantation. In particular, the expansion of oil palm plantation in Indonesia requires a lot of scientific data, information and knowledge on environmental settings, biodiversity & ecosystem services as well as human dimension aspects in order to determine the trade off between conservation and sustainable production, including the innovative ways to improve the socio-ecological performances of the system. This manuscript has discussed important findings that are useful for development

planners and other relevant stakeholders in order to define the minimum coverage of forested areas and also manage further development of the existing land uses (i.e. crop plantations) that are usually already stable or unchanged. The findings may be used as baseline to model the future land use development in other areas where forest cover is still changing. We suggest to consider the use of landscape approaches within the context of sustainable watershed management in order to take into account other important ecosystem services such as production, information, habitat etc.

---

## Short Comment (SC5) · 4 Jul 2017

Minimum forest cover for sustainable water flow regulation in a watershed under rapid expansion of oil palm and rubber plantations -The manuscript presents important results in determining a minimum forest cover within a watershed for sustainable water flow regulation. The research findings are useful for regional planning in order to prevent further expansion of plantation such as oil palm. Rapid expansion of palm oil or other mono-species plantation has reduced percentage of forest cover area and changed environmental condition, such as declining biodiversity and unbalance water flow. As the consequent of unbalance water flow leads to flooding in rainy season

and drought in dry season. -Although the manuscript explains important findings, however, it needs some improvement before publishing it in the international Journal. It will be useful if the accuracy of the model presented in the manuscript. It can be conducted by calculating the average deviation of the predicted and measured/observed variables using absolute values. Another method by graphing a scatter plot between the predicted and the observed variables, then graph a one-one line to show whether the predicted values above or under observed values (see example below).

-The other essential aspect is the continuation of lowflow in the dry period. In this manuscript the authors only explained the relation between % forest cover and BFI (baseflow/total flow). It is important to present lowflow, because there is an controversial issue regarding expansion of mono species plantation which is considered high water consumption and therefore it will be drought in a watershed when planted by mono species such oil palm. -Improvement of paragraph should be conducted, for example the second paragraph is too long. -Improvement of figures will present the manuscript better.

---

## Author Comment (AC1) · 12 Jul 2017

The requirement to keep a proportion of forest area of 30% in the entire watershed area is already highlighted in some of Indonesia's regulations. But, there is still a lack of scientific background on how to determine this percentage. To the best of our knowledge, this our manuscript is the first paper dealing with the scientific background of such regulations.

---

## Author Comment (AC2) · 12 Jul 2017

It is true that besides the land cover, the waterflow regulation in a watershed is also influenced by other factors such as soil type. Soils in our study area are dominated (80%) by two soil types, namely Tropodult and Dystropept (see Figure 1). In term of soil physical input parameter for SWAT model, both soils are rather similar. Therefore, impact of soil variability can be neglected in the study area.

In terms of rainfall variability, it is true that Indonesia has different rainfall patterns across the country. The western part of Indonesia, including Sumatra and Kalimantan are characterized by high annual rainfall. Meanwhile, the eastern part like Sulawesi,

East and West Java have less annual rainfall. Oil palm plantations are normally developed in areas with high amounts of annual rainfall like Sumatra and Kalimantan. Our study represents this type of bio-physical environment as stated in the title.

We agree to the comment that the evapotranspiration of similar tree species in several locations are not exactly the same. But the influence of this variability on the waterflow regulation of a watershed is relatively minor. The reason is that the potential evapotranspiration especially in the western part of Indonesia rarely deviates much from 4 - 6 mm day-1 due to the high humidity. In our study, the influence of reduced soil water infiltration is far more important than that of the evapotranspiration variability.
* * *
[Figure]

Figure 1. Soil type in BH and MT watersheds. Soil types respresent Fluvaquents (S-3), Humitropepts (S-4), Paleudults (S-7), Tropofluvents (S-10), Troposaprist (S-11), Tropodults (S-12), Dystrandepts (S-13), Dystropepts (S-14).

**Fig. 1.**

---

## Author Comment (AC3) · 12 Jul 2017

To obtain representative variability of land cover and land-use type in different watersheds, in our study, we selected two watersheds with different distributions of land cover (Figure 1 and 2), but with relatively similar soil types (Figure 3).

Concerning the suggestion to apply the SWAT model also in other areas, we believe that our study area is representative for most areas with oil palm development in terms of soil types, topography, and rainfall variability (please see our comment to Mr Ginting). Similar un-published studies in other watersheds using the SWAT model have been conducted by other researchers and they show trends similar to our findings.

[Figure]

[Figure]

[Figure]

Figure 1. Land cover in BH watershed (copied from Figure 3 of the manuscript)

**Fig. 1.**

[Figure]

Figure 2. Land cover in MT watershed (copied from Figure 3 of the manuscript)

**Fig. 2.**

[Figure]

Figure 3. Soil type in BH and MT watersheds. Soil types respresent Fluvaquents (S-3), Humitropepts (S-4), Paleudults (S-7), Tropofluvents (S-10), Troposaprist (S-11), Tropodults (S-12), Dystrandepts (S-13), Dystropepts (S-14).

**Fig. 3.**

---

## Author Comment (AC4) · 12 Jul 2017

Concerning the comment that the finding is particularly suitable to be implemented in the area where forest cover is still changing, we want to add that the finding is also useful as a guideline for afforestation programs of degraded watersheds where all forest is gone and also as a guideline to limit the proportion of oil palm area in a watershed where rapid oil palm expansion is going on.

---

## Author Comment (AC5) · 12 Jul 2017

The accuracy of the model has been stated in the manuscript. We use NSE to denote the accuracy of the model. The model gave satisfactory performance with the NSE values of 0.88 and 0.86 for calibration and 0.85 and 0.84 for validation in BH and MT watersheds respectively.

Concerning comment on lowflow, the BFI value already describes the low flow. If BFI value is small then baseflow during dry season (low flow) is also low.

We are thankful for the suggestion to improve Figures and writing style.

---

## Short Comment (SC6) · 14 Jul 2017

The area under study, the Jambi province of Sumatra, is experiencing a rapid decline in forest cover which is accompanied by a loss of a wide range of ecosystem functions. Local people in the area frequently report about an increasing severity of water scarcity in dry season as well as an increase of flooding events in rainy season. The latter is of greatest concern as flood events are increasingly impeding effective land use of local farmers. These observations are oftentimes related to the land use change and deforestation activities in recent years. Ongoing discussions about the linkages between forest transition and hydrological outcomes have pointed to the regional heterogeneity

and site specific characteristics of alterations in water flows. It is for this reason that empirical studies as well as site-specific hydrological models as those by Tarigan et al. are urgently needed to better understand the linkages between forest cover and water flows. In addition to discussions on the minimum amount of forest cover, I hope that further research will 1. give more insights to the question of where forest areas should be protected most urgently and 2. how we can implement policies of minimum forest cover without excluding vulnerable social groups from the possibilities to access and benefit from their natural resources.

---

## Short Comment (SC7) · 18 Jul 2017

Good understanding on the proportion of forest cover and the proportion of plantation cover are highly needed for maintaining a sustainable ecosystem service of water flow regulation in a watershed. The article has provide the valuable info by simulating some data in to a model, the SWAT hydrological model. However, several consideration should be taken into account, especially for spatial planners, before applying the proportions (the results) into the real field condition as validation of the model has never been performed (no explanation in the abstract). Furthermore, simplification or limitation consisted in the model has to be explained in the first place, so by doing that

people can measure properly the model results and can be used as a guide in the field.

Best regards, Fuad Nurdiansyah

---

## Referee Comment (RC1) · Anonymous Referee #1 · 25 Jul 2017

The manuscript deals with the effect of land cover and/or land use on a watershed response functioning. The authors investigated the influence of forest and monoculture plantations (oil palm and rubber plantations) on rainfall partitioning to direct runoff and subsurface flow for a humid tropical watershed in Indonesia. The results are based on streamflow as simulated by a calibrated SWAT model and observations across several watersheds and subsequently derived the direct runoff coefficient (C) and the baseflow index (BFI). The study exhibits a statistically significant correlation of percentage of forest covers in a watershed with C (negatively) and BFI (positively). On the other hand, the rubber and oil palm plantations showed flow regulation behaviour contrary to forest covers. Finally the study suggests the minimum forest cover requirement in

the study area (i.e. 30%) for sustainable ecosystem services. The topic is of current scientific interest and several studies have also investigated previously. However, the manuscript requires a substantial improvement of the methodology and, results and discussion to be publishable. Furthermore, the manuscript would benefit a lot with the inclusion of more discussions in the introduction section from previous similar studies in the tropical regions.

General comments 1. Given the previous several studies on the effect of land cover/use conversion on the hydrology of a watershed, the introductory section needs further literature review in this regard. It should also highlight the new contribution of this manuscript. 2. I think the organization of the methods section, in general, requires restructuring and further information. For example, there is no section that describes the general SWAT model and the SWAT model for the study area, which are important for general readers and non-SWAT users. 3. Section 3.2 and section 3.3 should be presented before section 3.1. Logically thinking, observation based model evaluations should be presented first and then results of analyses based on the model simulation. 4. No information is provided in the manuscript about the SWAT parameters, particularly the ones that control the surface runoff and the baseflow process. I think information about some of the sensitive parameters would give a good discussion points on the flow regulation behahavoir of different Landover/use in the study area. What was your observation on the calibrated SWAT parameters such as CN2, SOL_AWC, ALPHA_BF and CANMX among other? 5. The calibration and validation strategy are not clearly stated, albeit its importance in interpreting simulation outputs from SWAT. The calibration and validation period need to be explicitly stated. Which automatic calibration algorithm was used in SWAT-CUP? It is also essential use multiple evaluation criteria. 6. I encourage the authors to explicitly discuss the SWAT model simulation results are mainly arising due to changes in land cover not by wrong parameterization. SWAT is a highly parametrized model, therefore we might get the expected patterns for the wrong reason. This could be addressed by referring the calibrated SWAT parameters.

Specific comments Lines 1-2: I suggest to check the title. i) Since it is an application in tropical region in Indonesia, it needs to be specific. ii) It seems to me some action words are missing. You could simply add, for instance, "requirement" that reads as "Minimum forest cover requirement for sustainable water flow regulation: A case study in a watershed under rapid expansion of oil palm and rubber plantations in Indonesia" Lines 9-32: The abstract could be shortened to a certain extent by reducing the seemingly redundant sentences on flow regulation functioning and benefits, keep the most important points only. Lines 14-15: It is a bit confusing sentence, please improve the language. Line 40 " Lele, 200" please add 0 Line 40 "Functional water flow regulation reduces flood peaks by moderating direct runoff." It would be nice to add some references here. Line: 46:"base flow" remove space Line 46: ")]" remove the square bracket Lines: 69-71: Please improve the language Lines 72-73: Improve the language, for instance, "Distributed hydrologic models such as the Soil and Water Assessment Tool (SWAT) are useful to understand the effects of land use changes on watershed flow regulation. . . . . .." Lines 80-81: ". . . . . . is the direct runoff ratio of to rainfall." should be "is the ratio of direct runoff to rainfall" Line 88: Please add the size of the study area and perhaps the location coordinates. Lines 88-93: It would be informative to add information on the historical land cover change in the study area. Lines 94-98. I think the methodology description should not be included in study site description. I suggest to move this part to appropriate subsection in the methodology. Line 99: Replace "&" with and Line 101: "C & BFI" it should be "C and BFI" like in the abstract section and it should be consistent throughout the manuscript. Lines 102-104: Please improve the language. And it is somewhat similar with Lines 109-110 Lines 104-109: This is confusing! This describes the general SWAT model and I would rather expect a separate subsection for it. This should also tell how SWAT computes surface runoff, baseflow. . ...See the comments in the general comment. Line 114: I would prefer the areas in km2. Line 118-121: Describes the SWAT model setup for the study area. Therefore, I would expect to get this information before describing section 2.2 (Simulated C and BFI) values. Line 122: Add SWAT-CUP reference Lines 121-129: This

part tries to elaborate the model calibration and evaluation part. SWAT-CUP provides several options for model calibration, which one did you use in this study? Please be specific. When is your calibration and validation periods? I suggest separate subsection for model calibration and evaluation approach. Line 128: As demonstrated in several studies, NSE is sensitive to peak flows. You calibrated and evaluated your model using only NSE. How do you justify this? I think it would be good to add a few more performance indices in the evaluation so that the reader would have a better feel on the reliability of the model simulation outputs. Line 130: Again "&" remove throughout the manuscript. Line 158 "didn't" should be "did not" Lines 162-163 repetition see line 121 Line 163-164: Add more statistics Lines 165-167: what did you obtain from the comparison? How much they agree? What statistical measures did you use? Lines 168-173: More suitable in the methodology section. Lines 180-184 Too long sentence, it is better to follow simple sentences. Improve the language as well. Line 182: Oil palm harvest and oil palm circle are equal (i.e. 3 cm h-1). Lines 185-188: I'm puzzled by this conclusion. Is the rainfall distribution similar throughout the basin? Because if there is a spatial variation in rainfall magnitude, the effects of forest conversion on the flow regulation would vary accordingly. In Figure 4a, I see a C value less than 0.35 for forest cover about 20%, what do you think about this? Line 207: please improve the language Lines 207-214. I think this need more discussion. SWAT has a known limitations in simulating the low flow regime and that would have an effect on the BFI, as also mentioned by the authors. See the recent study for further discussion: Pfannerstill, M., B. Guse, and N. Fohrer, 2014a. A Multi-Storage Groundwater Concept for the SWAT Model to Emphasize Non- linear Groundwater Dynamics in Lowland Catchments. Hydro- logical Processes 28:5599-5612, DOI: 10.1002/hyp.10062

Line 344: ". . .MT(b,. . ." Line 376: Table 3, In MT watershed sub.wat.nr 23 has a 100% forest cover but the BFI is low, meaning low baseflow contribution from the groundwater. Justify this in the discussion. Line 379 Table 4, Please recheck the numbers and the calculations.

---

## Short Comment (SC8) · 25 Jul 2017

The damage of hydrological conditions of watersheds as a result of uncontrolled expansion of cultivated land and residential areas without applying land and water conservation principles are often the causes of erosion, sedimentation, low land productivity, land degradation and flood in Indonesia.

The efforts to harmonize the needs of land, agricultural development and regional development regards on environmental carrying capacity have been established by the Republic Government of Indonesia through making and the review of Provincial and District Spatial Plans (RTRWP/K). However this has been not able yet to address the

optimal need for forest to maintain and conserve water flow from watershed.

This study shows that 30% of minimum forest proportion is capable to produce a flow coefficient value less than 0.28. The use of SWAT hydrology method to calculate the minimum forest cover requirement in watershed is a new technology application in Indonesia. This innovation will be useful for the Ministry of the Environment and Forestry and other government agencies to determine the proportion of land uses for forests, plantations, build up areas and others in watershed area.

The minimum adequacy proportion for forest land cover in Indonesia has been regulated in Law no. 41 of 1999 on Forestry, Law no. 26 of 2007 on Regional Planning, Government Regulation of the Republic of Indonesia No. 44 of 2004 on Forestry Planning, and Minister of Home Affairs Regulation no. 67 of 2012 on Guidelines for Implementation of Strategic Environmental Assessment in the preparation or evaluation of regional development plans. This research can reinforce the legislation by recommending a minimum proportion for forest cover at 30% and a maximum closure for plantations at 40% within the watershed.

---

## Referee Comment (RC2) · Anonymous Referee #2 · 28 Jul 2017

General comments:

I found the topic and results described in this manuscript to be quite interesting. There is very limited information available in the literature to date regarding the potential effects of expanded production of rubber or oil palm trees, using SWAT model or any other modeling approach. Thus I think that the information reported in this manuscript will ultimately prove to be a useful contribution to Hydrology and Earth System Sciences (HESS) and the existing literature in general. However, I believe that the current manuscript suffers from several deficiencies including inadequate review of existing literature, insufficient description of SWAT and key input parameters (including coeffi-

cients used for rubber tree and oil palm tree in the crop parameter file), lack of in-depth description of SWAT calibration and validation results, and an inadequate description of the simulated watersheds. Specific comments regarding these issues are provided below.

Specific comments

1) Abstract: The Abstract needs to be considerably revised to reflect more of the actual quantitative results of the study versus the "general discussion" that dominates much of the abstract between lines 9 to 24. The revised abstract should include a summary of the baseline calibration and validation results.

2) Lines 43-45: I would suggest you rewrite this sentence to read something like: "This vertical movement of water in the soil determines how much water flows as direct runoff and how much percolates to the water table where it sustains baseflow or groundwater (references)."

3) Lines 49-68: Please include citation and discussion of some "big picture" studies regarding the impacts of Palm Oil and/or Rubber Trees in the southeast Asia region such as those listed immediately below.

Mukherjeea, I. & B.K. Sovacoo. 2014. Palm oil-based biofuels and sustainability in southeast Asia: A review of Indonesia, Malaysia, and Thailand. 37: 1-12. DOI: 10.1016/j.rser.2014.05.001.

Wilcove et al. 2013. Navjot's nightmare revisited: logging, agriculture, and biodiversity in Southeast Asia. Trends in Ecology and Evolution 28(9): 531-540. DOI: 10.1016/j.tree.2013.04.005.

Ziegler et al. 2009. The Rubber Juggernaut. Science 324: 1024–1025. DOI: 10.1126/science.1173833.

Ziegler et al. 2011. Recognizing Contemporary Roles of Swidden Agriculture in Transforming Landscapes of Southeast Asia. Conservation Biology 25(4): 846-848. Available at: http://www.jstor.org/stable/27976544.

4) Lines 69-70: Please expand this discussion to provide a broader review of different modeling and other analysis methods, beyond the option of SWAT, available to assess the impacts of expanded rubber and oil palm plantations in the southeast Asia region.

5) The expanded paragraph noted in comment 3 should be followed by a specific paragraph about SWAT including relevant review studies about SWAT and a more in-depth review of how SWAT has been used for land use change analyses. Note that the Zhang et al. (2013) article you cite in line 76 is not a very good choice regarding reviews of SWAT studies; please instead cite one or more of the studies listed on the webpage at http://swat.tamu.edu/publications/special-issues/ or in the "SWAT Publications box" in http://swat.tamu.edu/. Please also cite some relevant SWAT "land use change studies" (see the SWAT Literature Database that can again be accessed on the SWAT model homepage) such as those listed here:

Babel, M.S., B. Shrestha and S.R. Perret. 2011. Hydrological impact of biofuel production: A case study of the Khlong Phlo Watershed in Thailand. Agricultural Water Management. 101(1): 8-26. DOI: 10.1016/j.agwat.2011.08.019.

Marhaento et al. 2017. Attribution of changes in the water balance of a tropical catchment to land use change using the SWAT model. Hydrological Processes. 31(11): 2029–2040. DOI: 10.1002/hyp.11167.

Tan et al. 2015. Impacts of land-use and climate variability on hydrological components in the Johor River basin, Malaysia. Hydrological Sciences Journal. 60(5): 873-889. DOI: 10.1080/02626667.2014.967246.

Tarigan et al. 2016. Mitigation options for improving the ecosystem function of water flow regulation in a watershed with rapid expansion of oil palm plantations. Sustainability of Water Quality and Ecology . 8: 4-13. DOI: 10.1016/j.swaqe.2016.05.001.

Wangpimool et al. 2017. The impact of Para rubber expansion on streamflow and

other water balance components of the Nam Loei River Basin, Thailand. Water. 9(1) DOI: 10.3390/w9010001.

6) Lines 71-73: These two current sentences have grammatical problems. As a part of comment 4, I suggest that you revise the text as follows: "A useful tool to answer this question is the Soil and Water Assessment Tool (SWAT) ecohydrological model (Arnold et al., 1998; 2012), which quantifies the water balance of a watershed on a daily basis (Neitsch et al., 2009) and has been recommended for the evaluation of hydrological ecosystem services of a watershed (Vigerstol et al., 2011)."

7) Study area description: The two study watersheds should be described in depth in this subsection rather than being referenced later in subsection 2.2 (please describe the area of the watersheds in km2 rather than ha). More detailed land use information (percentages of each type of land use) for the two watersheds should be provided (rather than waiting until subsections 2.3.1 and 3.2 to describe some of that information), as well as more information about the natural vegetation, and rubber and oil palm plantations (growth cycles, management practices, time period of plantation development, etc.). Further details about the typical porosity and other characteristics of the soils in the study watersheds would also be useful.

8) In relation to comment 6, some description of all six macro watersheds shown in Figure 1 should also be provided in the Study area description subsection. Who defined these six watersheds and why? It is clear that hydrologic data was collected for the watersheds but the current text is vague regarding the overall purpose of these six watersheds.

9) Also in relation to comment 6, please describe the "small watersheds" referenced in lines 144-145 and 195-196 and shown in Figure 1 in the study area subsection, rather than waiting to describe those in current section 2.3.1 (and that information does not need to be repeated at the start of section 3.2). What other hydrologic data were collected for those small watersheds besides the C values?

10) Please rewrite "C&BFI" as "C and BFI" throughout the text.

11) A SWAT Description subsection needs to be added to the manuscript. This should note the specific version of the model used for the study (including the Revision number) and provide a succinct overview of the model, especially regarding components that were particularly important for the study you conducted. A description of the crop parameters used for the rubber and oil palm trees, and other vegetation in the watersehds, should also be provided (those parameters could be described later in the methods if more appropriate). See the Wangpimool et al. article listed in comment 4 above regarding revised rubber tree crop parameters they used in their study.

12) The information in current subsection 2.2 needs to be revised to present key aspects of your methods in a more coherent manner. Some suggestions:

a) A separate subsection is needed describing the various input data used in the study (including citation of Table 1), which could be a part of the expanded study area description. More complete citations of the input data sources are needed. Information such as the number of subwatersheds delineated for each watershed (and the number of HRUs used is also pertinent) should be discussed in this subsection rather than at the start of the Results section. Consider moving the information in section 5 to this subsection.

b) An expanded description of the SWAT calibration and validation procedures is needed, which again should be in a separate subsection. This should include a description of the calibration parameters used in the study, including the default value (or initial value range) and the final calibrated values. Please also provide a description of any sensitivity analyses that was performed and provide a description of the specific baseflow separation techniques that were used in the calibration process. A description of measured baseflow data, or proxy baseflow data obtained via literature sources or expert opinion, is also important in relation to the use of the BFI indicator in your study.

c) I suggest you then introduce a third subsection that describes the specific C and BFI methods that were used in your analyses.

13) Please expand on your discussion of the calibration and validation results. This should include showing hydrograph comparisons between the simulated and measured outputs and discussion of your results in the context of model evaluation criteria suggested in the two Moriasi et al. studies listed here:

Moriasi et al. 2007. Model evaluation guidelines for systematic quantification of accuracy in watershed simulations. Transactions of the ASABE. 50(3): 885-900. Doi: 10.13031/2013.23153.

Moriasi et al. 2015. Hydrologic and water quality models: performance measures and evaluation criteria. Transactions of the ASABE. 58(6): 1763-1785. Doi: 10.13031/trans.58.10715.

14) Line 131: I think the word "was" should be "were". Why were simulated values that were within an "order of magnitude" of the measured values considered acceptable? It appears that the average measured and simulated C values reported in Tables 5 versus 6 were almost identical; that would indicate that the "order of magnitude" criteria is unnecessary?

15) Sentence in lines 184-185: The phrase "as acceptable for a good watershed service" in this sentence sounds odd. A suggested revision is: "The Ministry of Forestry of Indonesia considers C values < 0.35 to be adequate to support required ecosystem services for Indonesian watersheds (citation)."

16) Conclusions: Some expansion of your Conclusions section is warranted. Please include additional quantitative information from both the baseline testing results as well as the C and BFI analyses.
* * *

---

## Author Comment (AC6) · 30 Jul 2017

Regarding to the proposed further research questions:

1) Where forest areas should be protected most urgently

In term of hydrological functions, the main role of forest is to increase water recharge. Increased water recharge will further reduce flooding during wet season and increase baseflow during dry season. For water recharging purposes, protected forest area should be situated at the upper part of each sub-watershed normally characterized by steeper slope and lighter soil texture.

[Figure]

2) How we can implement policies of minimum forest cover without excluding vulnerable social groups from the possibilities to access and benefit from their natural resources.

This is an integrated research question requiring contribution from ecological, social and economic aspects.

---

## Author Comment (AC7) · 30 Jul 2017

There are at least two approaches to construct relation between the proportion of the land use/cover and the hydrological characteristics in a watershed scale, namely empirical and distributed hydrological model approaches. Empirical approach requires long-term discharge data for each of the 48 sub-watersheds in our study area. These data are lacking particularly in our study area. The distributed hydrological model approach uses sets of physical-hydrological laws and collected bio-physical parameters such as soil, topography and land use/cover for model inputs. We used the hydrological model approach in our study. We validated the model results following standard procedures for the calibration and validation processes of the SWAT hydrological model. In addition, we also carried out some field experiment and data collection to validate the C and BFI values as it was explained in the manuscript.

---

## Author Comment (AC8) · 30 Jul 2017

We agree that the Ministry of the Environment and Forestry and other government agencies needs scientific reference to recommend particular proportion of forest cover in a watershed. The scientific reference was still lacking and we hope that this paper can serve this need. We are thankful for additional information on the related government regulations pertaining to the topic of the manuscript.

---

## Author Comment (AC9) · 22 Aug 2017

Author responses to Anonymous Referee #1

The manuscript deals with the effect of land cover and/or land use on a watershed response functioning. The authors investigated the influence of forest and monoculture plantations (oil palm and rubber plantations) on rainfall partitioning to direct runoff and subsurface flow for a humid tropical watershed in Indonesia. The results are based on streamflow as simulated by a calibrated SWAT model and observations across several watersheds and subsequently derived the direct runoff coefficient (C) and the baseflow index (BFI). The study exhibits a statistically significant correlation of percentage of

forest covers in a watershed with C (negatively) and BFI (positively). On the other hand, the rubber and oil palm plantations showed flow regulation behavior contrary to forest covers. Finally the study suggests the minimum forest cover requirement in the study area (i.e. 30%) for sustainable ecosystem services. The topic is of current scientific interest and several studies have also investigated previously. However, the manuscript requires a substantial improvement of the methodology and, results and discussion to be publishable. Furthermore, the manuscript would benefit a lot with the inclusion of more discussions in the introduction section from previous similar studies in the tropical regions.
* * *
General comments

1. Given the previous several studies on the effect of land cover/use conversion on the hydrology of a watershed, the introductory section needs further literature review in this regard. It should also highlight the new contribution of this manuscript.

We appreciate the referee's suggestions, and have added more literature review on the effect of land cover/use conversion on the hydrology of a watershed in the introductory section. (See the description in the following paragraphs)

Line 53-58 (Oil palm and local water cycle) The impact of tropical rainforest conversion into plantations such as oil palm and rubber is not limited to the biodiversity loss, decreased carbon stock, and increased greenhouse gas emissions but also change local water cycle including increased transpiration (Roell et al., 2015; Hardanto et al., 2017), increased evapotranspiration (Babel et al., 2011; Meijide et al., 2017), decreased infiltration (Banabas et al., 2008; Tarigan, et al., 2016), increased flooding (Tarigan, 2016), decreased low flow (Yusop et al., 2007; Adnan and Atkinson, 2011; Comte et al., 2012; Merten et al., 2016) and water quality (Babel et al., 2011).The change of the water cycle will in turn affect water flow regulation function of a watershed.

Line 93-102 (Land use change and the SWAT model) Marhaento et al. (2017) used the SWAT model to simulate impact of forest cover and agriculture land use on the runoff coefficient and the ratio of base flow to stream flow in Java Island Indonesia and found that forest cover change from 48.7% to 16.9% resulted in the increased of the runoff coefficient (C) to 44.6% and decrease of the ratio of base flow to stream flow to 31.1% showing similar trend with that of our results. Meanwhile, Wangpimool et al., (2017) found that annual reduction of about 3% in the basin average water yield based on the SWAT model simulation due to the rubber expansion in Thailand from 2002 to 2009. Babel et al., (2011) simulated impact of oil palm expansion using SWAT in Thailand and reported increased nitrate loading (1.3 to 51.7%) to the surface water. The new contribution of our study is the establishment of quantitative relation between forest cover and flow indicators in a watershed, which can be used as a guide for spatial planners to determine the minimum proportion of forest conservation area to maintain a sustainable ecosystem service of water flow regulation in a watershed. Tarigan et al. (2016) used SWAT model to simulate impact of soil and water conservation practices on low flow in oil palm dominated watersheds in Jambi Provinces, Indonesia dominated watersheds in Jambi Provinces, Indonesia

Some newly added References:

Adnan, N. A., Atkinson, P. M. 2011. Exploring the impact of climate and land use changes on streamflow trends in a monsoon catchment. International Journal of Climatology 31, 815–831.

Babel, M.S., B. Shrestha and S.R. Perret. 2011. Hydrological impact of biofuel production: A case study of the Khlong Phlo Watershed in Thailand. Agricultural Water Management. 101(1): 8-26. DOI: 10.1016/j.agwat.2011.08.019.

Comte, I., Colin, F., Whalen, J.K., Gruenberger, O., Calliman, J.P., 2012. Agricultural Practices in Oil Palm Plantations and Their Impact on Hydrological Changes, Nutrient Fluxes and Water Quality in Indonesia: A Review. Advances in Agronomy, Volume 116,

2012 Elsevier Inc. ISSN 0065-2113.

Hardanto A, Röll A, Furong N, Meijide A, Hendrayanto, Hölscher D (2017) Oil palm and rubber tree water use patterns - effects of topography and flooding. Frontiers in Plant Science 8: 452 http://journal.frontiersin.org/article/10.3389/fpls.2017.00452/full

Hardanto A, Röll A, Hendrayanto, Hölscher D (2017) Tree soil water uptake and transpiration in mono-cultural and jungle rubber stands of Sumatra. Forest Ecology and Management 397: 67-77 http://www.sciencedirect.com/science/article/pii/S0378112717304747?via%3Dihub

Marhaento et al. 2017. Attribution of changes in the water balance of a tropical catchment to land use change using the SWAT model. Hydrological Processes. 31(11):2029–2040. DOI: 10.1002/hyp.11167.

Meijide A, Röll A, Fan Y, Herbst M, Niu F, Tiedemann F, June T, Rauf A, Hölscher D, Knohl A (2017) Controls of water and energy fluxes in oil palm plantations: Environmental variables and oil palm age. Agricultural and Forest Meteorology 239: 71-85 http://www.sciencedirect.com/science/article/pii/S0168192317300771

Röll A, Niu FR, Meijide A, Hardanto A, Hendrayanto, Knohl A, Hölscher D (2015) Transpiration in an oil palm landscape: effects of palm age. Biogeosciences 12: 9209-9242. http://www.biogeosciences-discuss.net/12/9209/2015/bgd-12-9209-2015-print.pdf

Wangpimool et al. 2017. The impact of Para rubber expansion on streamflow and other water balance components of the Nam Loei River Basin, Thailand. Water. 9(1) DOI: 10.3390/w9010001.

Van Griensven, A., Maharjan, S., & Alemayehu, T. (2014). Improved simulation of evapotranspiration for land use and climate change impact analysis at catchment scale. International Environmental Modelling and Software Society (iEMSs) 7th International Congress on Environmental Modelling and Software.

———————————————————————————

2. I think the organization of the methods section, in general, requires restructuring and further information. For example, there is no section that describes the general SWAT model and the SWAT model for the study area, which are important for general readers and non-SWAT users.

We agree with the referee's suggestions, and have re-structured the method section as follows:

2. Methods

2.1 Study area

2.1.1. Land use and soil characteristics

2.1.2 Watershed characteristics

1. Macro watersheds 2. Small watersheds

2.2 Flow simulation

2.2.1 SWAT model

1. Crop and soil parameters 2. Input data 3. Model validation and calibration

2.2.2 Simulated C and BFI values

2.2.3 Observed C and BFI values

1. Observed C values 2. Observed BFI values

The general SWAT model and the SWAT model for the study area are described in Subsection 2.2.1. (See the description in the following paragraphs)

2.2.1 SWAT model We used the SWAT model version 2012 (Arnold et al., 2012). The SWAT model is a continuous model, i.e. a long-term yield model. The model was developed to simulate the impact of land cover/management practices on the stream-flow in complex watersheds with varying soil, land use and management condition over

long periods of time. Major model components include weather, hydrology, soil temperature and properties, plant growth, nutrients, and land management (Arnold et al., 2012; Neitsch et al., 2009). Delineation of watersheds and their sub-watersheds in our study area was carried out automatically by the SWAT model and was based on a digital elevation model (DEM) with a 30-m resolution. During the automatic delineation we pre-defined an area of 50.000 ha as a threshold for a minimum sub-watershed area. Based on this threshold, both study watersheds in our study area were further sub-divided into 25 and 23 sub-watersheds, respectively. The sub-watershed is further sub-divided into hydrological response units (HRUs) with homogeneous hydrological unit defined by topography, soil, and land use characteristics. Hydrological outputs are then calculated in the HRUs based on the water balance equation. Output of the SWAT model include total stream flow, surface flow and base flow. These output were used to calculate the C and BFI values for each sub-watershed. For this simulation, the SWAT model required other inputs such as climate data, as well as soil and land-use maps for each sub-watershed (Table 1).

We also carried out field data collection including hydraulic conductivity (SOL_K), bulk density (SOL_BD), available water content (SOL_AWC) and texture for SWAT model input. Digital Elevation Model with 30 m pixel resolution is available from the National Aeronautics and Space Agency. Rainfall and climate date are available from the Meteorology and Geophysics Agency. The streamflow data of the six macro watersheds were provided by the Ministry for Public work. The land use data are available from the Regional Planning office. All these data are freely available for research purposes by official request to the corresponding institutions. The time series streamflow and the rainfall records for the small catchments, the resampled soil hydraulic conductivity, bulk density, available water content and texture are deposited by the first author office at Bogor Agricultural University and EFForTS Database (https://efforts-is.uni-goettingen.de). The land-use and soil map for the study area was obtained from Jambi Province Regional Planning (BAPEDA, 2013) and Agricultural Plantation offices (Ditjenbun, 2013). Soils in our study area are dominated by two soil types, namely

[Figure]

Tropodult and Dystropept (Figure 1).

———————————————————————————————————-

3. Section 3.2 and section 3.3 should be presented before section 3.1. Logically thinking, observation based model evaluations should be presented first and then results of analyses based on the model simulation.

We agree with the referee's suggestions, and have re-structured the discussion section as follows:

3. Results and Discussion

3.1 Performance of the SWAT model

3.2 Observed C and BFI values

3.3 Simulated C and BFI values

3.4 Correlation of percentage of forest covers in a watershed with C and BFI

3.5 Application of the research result

———————————————————————————————————

4. No information is provided in the manuscript about the SWAT parameters, particularly the ones that control the surface runoff and the baseflow process. I think information about some of the sensitive parameters would give a good discussion points on the flow regulation behavior of different Landover/use in the study area. What was your observation on the calibrated SWAT parameters such as CN2, SOL_AWC, ALPHA_BF and CANMX among other?

We agree with the referee's suggestions, and have added detail information about the SWAT parameters such as CANMX and CN2 in subsection 2.2.1 of the method section (see the description in the following paragraphs).

According to Griensven et al. (2014), the SWAT is designed for temperate regions so

Interactive
comment

that it is necessary to adapt the crop parameters for application in a tropical region. In this respect, we adjusted the crop parameters, directly related to the flow component such as CANMX and CN. To adapt these values we carried out field measurement on several important hydrological component including interception, infiltration, and overland flow (Figure 2).

Canopy Storage (CANMX)

Interception reduces the amount of water reaching the ground and consequently reduces streamflow. We measured interception in oil palm, rubber, agroforest, and forest trees at the plot between November 2012 and February 2013. In total there were 30 rainfall events during this time, representing light to heavy rain. In oil palm, rainfall is not only intercepted by leaves and branches but also by hollow spaces between fronds and trunk. This type of interception is called trunk storage and may have led to the slightly increased interception in oil palm. Interception in oil palm was rather similar to interception in the forest. The measured interception (Figure 3) values were used as an estimate of (CANMX), which serves as an input parameter for the SWAT model. The CANMX is the maximum amount of water that can be trapped in the canopy and trunks when they are fully developed. Higher CANMX values reduce potential runoff during heavy rains. Beside CANMX we also adapted other crop parameters such as OV_N, BLAI. CHTMX, T_BASE and T_OPT (Table 2).

Adapted CN values for oil palm and rubber land uses.

One important parameter of the SWAT model related to surface runoff modeling is SCS curve number (CN, Arnold et al., 2012). It determines proportion of rainfall becoming surface runoff. Its value range from 0-100. The bigger the value the higher the proportion of surface run of on a particular rainfall event. The SCS curve number (CN) is differentiated into Hydrologic Soil Groups (HSG) A,B,C, and D which are a function soil's infiltration. We measured soil infiltration and surface runoff in the typical land use types in our study area i.e. oil palm, rubber, and forest. Infiltration was measured

using a double-ring infiltrometer. No infiltration measurement was carried out under agroforest as infiltration in agroforest is likely similar to infiltration in secondary forest. Infiltration measurements in different land-use types from the study area showed the following order: oil palm harvest path (3 cm h-1) < rubber (7-7.8 cm h-1) < forest (47 cm h-1). The infiltration in the oil palm, rubber plantations were markedly lower than those at the forest.

The surface runoff in oil palm and rubber plantation were significantly higher than those in agroforest and forest (Figure 4). Low infiltration capacity in oil palm and rubber plantations was one reason for higher surface the plantation land use (Tarigan et al. 2016).

Due to the high surface runoff and the infiltration rate, we adopted HSG-D category for all HRUs in oil palm and rubber land uses irrespective of soil types (Table 3). For forest and agroforest, we assumed that the CN value was similar to those of forest evergreen (FRSE) and forest mixed (FRST) values in the SWAT crop database respectively.

——————————————————————————————

5. The calibration and validation strategy are not clearly stated, albeit its importance in interpreting simulation outputs from SWAT. The calibration and validation period need to be explicitly stated. Which automatic calibration algorithm was used in SWAT-CUP? It is also essential use multiple evaluation criteria.

We agree with the referee's suggestions, and have described in detail the calibration and validation strategy and period (see the description in the specific comment nr. 19 below) We calibrated the model using the Latin hypercube sampling approach from the Sequential Uncertainty Fitting version 2 in the SWAT Calibration and Uncertainty Procedure (SWAT‐CUP) package. First parameter ranges were determined based on minimum and maximum values allowed in SWAT. The SWAT‐CUP is an interface for auto-calibration that was developed for SWAT. The interface links any calibration/uncertainty or sensitivity program to SWAT (Abbaspour, 2015). The discharge

data of BH and MT watersheds used for calibration and validation were available for the period of 2005-2014. The calibration was carried out in year 2007-2009 and the validation in year 2012-2014. We evaluated the model using Nash-Sutcliff efficiency (NSE) and Percent Bias (PBIAS). The NSE is a normalized statistic that determines the relative magnitude of the residual variance ("noise") compared to the measured data variance ("information") (Nash and Sutcliffe, 1970). The PBIAS measures the average tendency of the simulated data to be larger or smaller than the observations Gupta et al., (1999). The optimum value is zero, and low magnitude values indicate better simulations. Positive values of PBIAS indicate model underestimation and negative values indicate model overestimation. The model input parameters that were used for the calibration process and their fitted values after calibration are shown in Table 4.

The ALPHA_BF (baseflow recession constant) was calculated from daily streamflow hydrograph plotted on semi-log paper.

——————————————————————————————————————

6. I encourage the authors to explicitly discuss the SWAT model simulation results are mainly arising due to changes in land cover not by wrong parameterization. SWAT is a highly parameterized model, therefore we might get the expected patterns for the wrong reason. This could be addressed by referring the calibrated SWAT parameters.

We appreciate very much the referee's concerns, and have explicitly discussed the SWAT model simulation to ensure that the results are mainly arising due to changes in land cover not by wrong parameterization. (See the description in the following paragraphs)

The CN value is the most sensitive parameter of the SWAT model . We realize that SWAT is designed for temperate regions so that it is necessary to adapt the crop parameters for SWAT model input in the tropical region (Van Griensven et al., 2014). To avoid wrong parameterization of the sensitive value we carried out the following steps: a) Adapting CN and CANMX values based on the field measurement on water cycle

related directly to the flow components including interception, infiltration, and surface runoff (see general comment 4 above), b) Replicating SWAT model simulation in two study watersheds, and c) Collecting time series streamflow data to calculate observed C and FBI and to get impression whether the C and BFI values calculated form the SWAT model really reflects the field observation (despite good performance of the model in our study).

————————————————————————————————————————————

Specific comments

1) Lines 1-2: I suggest to check the title. i) Since it is an application in tropical region in Indonesia, it needs to be specific. ii) It seems to me some action words are missing. You could simply add, for instance, "requirement" that reads as "Minimum forest cover requirement for sustainable water flow regulation: A case study in a watershed under rapid expansion of oil palm and rubber plantations in Indonesia"

We appreciate the referee's suggestion. The title has been changed to: "Minimum forest cover requirement for sustainable water flow regulation: A case study in a watershed under rapid expansion of oil palm and rubber plantations in Indonesia"

————————————————————————————————————————————

2) Lines 9-32: The abstract could be shortened to a certain extent by reducing the seemingly redundant sentences on flow regulation functioning and benefits, keep the most important points only.

We agree with the referee's suggestions, and have improved the abstract accordingly.

Abstract In many tropical regions, rapid expansion of monoculture plantations has led to a sharp decline of forest cover potentially degraded the water flow regulation function of watersheds. In a watershed where expansion of agricultural plantations occurs rapidly, the regional planner need to know the minimum proportion of forest cover required to maintain proper water flow regulation function of a watershed. Research dealing with

this issue is still rare, especially in the tropical area where oil palm expansion occurs at alarming rate. We investigated the impact of forest and monoculture plantations (oil palm and rubber plantations) on rainfall partitioning to direct runoff and subsurface flow for a humid tropical watershed in Indonesia. The results are based on streamflow as simulated by a calibrated SWAT model and observations across several watersheds and subsequently derived the direct runoff coefficient (C) and the baseflow index (BFI). The model gave satisfactory performance with the NSE values of 0.80-0.88 (baseline calibration) and 0.80 - 0.85 (validation); and the PBIAS values of -2.9 - 1.2 (calibration) and 7.0-11.9 (validation). The study exhibits a statistically significant correlation of percentage of forest covers in a watershed with C (negatively) and BFI (positively). On the other hand, the rubber and oil palm plantations showed flow regulation behavior contrary to forest covers. Finally the study suggests the minimum forest cover requirement in the study area (i.e. 30%) for sustainable ecosystem services. The new contribution of our study is the establishment of quantitative relation between forest cover and flow indicators in a watershed, which can be used as a guide for regional planners to determine the minimum proportion of forest conservation area to maintain a sustainable ecosystem service of water flow regulation in a watershed.

————————————————————————————————-

3) Lines 14-15: It is a bit confusing sentence, please improve the language.

The referee appears to be correct. We have removed the confusing sentence while shortening the abstract.

————————————————————————————————-

4) Line 40 " Lele, 200" please add 0

We thank the referee for the correction.

————————————————————————————————-

5) Line 40 "Functional water flow regulation reduces flood peaks by moderating direct

runoff." It would be nice to add some references here.

We have provided the relevant reference

————————————————————————————————————-

6) Line: 46:"base flow" remove space

Revision made; we inserted the space.

————————————————————————————————————-

7) Line 46: ")]" remove the square bracket

Revision made; we removed the square bracket

————————————————————————————————————-

8) Lines: 69-71: Please improve the language

We thank the referee for pointing this out; we have improved the language Line 69-71. As a consequence of reduced infiltration rate in the plantation areas, the surface runoff become higher promoting higher peak discharge. One alternative to reduce this impact is by maintaining sufficient proportion of the forested areas in the watershed promoting higher infiltration rate.

————————————————————————————————————-

9) Lines 72-73: Improve the language, for instance, "Distributed hydrologic models such as the Soil and Water Assessment Tool (SWAT) are useful to understand the effects of land use changes on watershed flow regulation: : :: : :." We thank the referee for pointing this out; we adapted the sentences suggested by the referees.

Line 72-73 Distributed hydrologic models such as the Soil and Water Assessment Tool (SWAT) ecohydrological model (Arnold et al., 1998; 2012) are useful to understand the effects of land use changes on watershed flow regulation

——————————————————————————————————————————-

10) Lines 80-81: ": : :: : : is the direct runoff ratio of to rainfall." should be "is the ratio of direct runoff to rainfall"

We thank the referee for the correction. We have revised the sentence.

——————————————————————————————————————————-

11) Line 88: Please add the size of the study area and perhaps the location coordinates.

We have added the size of the study area and the location coordinates.

Line 88-89 The study site covers an area of approximately 31 868 km2, is located at 1o54'31.4"S - 103o16'7.9" E in the Jambi Province of Sumatra (Fig. 1a).

——————————————————————————————————————————-

12) Lines 88-93: It would be informative to add information on the historical land cover change in the study area.

We thank the referee for the suggestion. We have added more information on the historical land cover change in the study area. (see the description in the following paragraphs)

The oil palm expansion in our study area (Jambi Province) increased almost 400% from 150,000 ha in 1996 to 600,000 ha in 2011 (Setiadi et al., 2011). The area under rubber increased from 500,000 to 650,000 ha in the same time period (Ditjenbun, 2013). In 2013, only 30% of Jambi Province was covered with rainforest (mainly located in mountainous areas), while 55% was already converted into agricultural land, and 10% of the land was degraded/fallow potentially converted to monoculture plantation (Drescher et al. 2016).

——————————————————————————————————————————-

13) Lines 94-98. I think the methodology description should not be included in study site description. I suggest to move this part to appropriate subsection in the methodology. We thank the referee for the suggestion. We have moved this part to method section.

——————————————————————————————————————-

14) Line 99: Replace "&" with and

Revision made; we replaced "&" with "and".

——————————————————————————————————————-

15) Line 101: "C & BFI" it should be "C and BFI" like in the abstract section and it should be consistent throughout the manuscript.

Revision made; we replaced "&" with "and"

——————————————————————————————————————-

16) Lines 102-104: Please improve the language. And it is somewhat similar with Lines 109-110

We thank the referee for pointing this out; we have removed the duplication.

——————————————————————————————————————-

17) Lines 104-109: This is confusing! This describes the general SWAT model and I would rather expect a separate subsection for it. This should also tell how SWAT computes surface runoff, baseflow: : :..See the comments in the general comment.

We thank the referee for the suggestion. We have added subsection 2.2.1 under which we have described the general SWAT model and the model setup in the general comment nr. 5.

2.2.1 SWAT model setup 1. Crop and Soil parameters 2. Input data 3. Model validation and calibration

[Figure]

———————————————————————————————————-

18) Line 114: I would prefer the areas in km2.

Revision made; we have replaced "ha" with "km2"

———————————————————————————————————-

19) Line 118-121: Describes the SWAT model setup for the study area. Therefore, I would expect to get this information before describing section 2.2 (Simulated C and BFI) values.

We thank the referee for the suggestion. We have added subsection 2.2.1 in the method section describing the SWAT model setup including input data, plantation-crop parameter and model validation and calibration. The subsections have been described in detail in the general comment 2, 4 and 5

2.2.1 SWAT model 1. Crop and Soil parameters 2. Input data 3. Model validation and calibration

———————————————————————————————————-

20) Line 122: Add SWAT-CUP reference

Revision made; we have added the SWAT-CUP reference. Line 122 The SWAT‐CUP is an interface for auto-calibration that was developed for SWAT model (Abbaspour et al., 2007, 2011, 2015).

———————————————————————————————————-

21) Lines 121-129: This part tries to elaborate the model calibration and evaluation part. SWAT-CUP provides several options for model calibration, which one did you use in this study? Please be specific. When is your calibration and validation periods? I suggest separate subsection for model calibration and evaluation approach.

We thank the referee for pointing this out. We have elaborated the model calibration

and validation in a new subsection (2.2.1) in the method section (see also general comment nr. 5). The program SUFI-2 and PBIAS in the SWAT-CUP software package were used for calibration and validation (.Abbaspour et al., 2011). The calibration and validation periods of the SWAT model were carried out in 2007-2009 and 2013-2014 respectively (Figure 5).
* * *
22) Line 128: As demonstrated in several studies, NSE is sensitive to peak flows. You calibrated and evaluated your model using only NSE. How do you justify this? I think it would be good to add a few more performance indices in the evaluation so that the reader would have a better feel on the reliability of the model simulation outputs.

We thank the referee for the suggestion. We have added one more indices, i.e. Percent bias (PBIAS) for the evaluation. Percent bias measures the average tendency of the simulated data to be larger or smaller than the observations. The optimum value is zero, where low magnitude values indicate better simulations. Positive values of PBIAS indicate model underestimation and negative values indicate model over estimation.

Line 163-167 Overall, the model performance was satisfactory with the NSE values of 0.80-0.88 (calibration) and 0.80 - 0.85 (validation); and the PBIAS values of -2.9 - 1.2, (calibration) and 7.0-11.9 (validation).
* * *
23) Line 130: Again "&" remove throughout the manuscript.

Correction made
* * *
24) Line 158 "didn't" should be "did not"

Correction made

———————————————————————————————————-

25) Lines 162-163 repetition see line 121

We thank the referee for pointing this out. We have removed the line 162-163

———————————————————————————————————-

26) Line 163-164: Add more statistics

Revision made; we have added more statistics, namely percent bias (PBIAS). The PBIAS measures the average tendency of the simulated data to be larger or smaller than the observations. The optimum value is zero, and low magnitude values indicate better simulations. Positive values of PBIAS indicate model underestimation and negative values indicate model overestimation.

———————————————————————————————————-

27) Lines 165-167: what did you obtain from the comparison? How much they agree? What statistical measures did you use?

We thank the referee for pointing this out. Actually, we made comparison in Line 198-202 for C value, and Line 201-214 for BFI value. The lines 165-167 was misplaced which have been moved to Subsection 2.2.3 (Line 135)

Line 198-202 To find out whether the simulated C values (Table 3) are comparable to the observed C values obtained from small watershed experiments (Table 4), we selected simulated C values from all sub-watersheds (Table 5) with a land cover proportions similar to those of the two observed small watersheds. The comparison showed that the average of the simulated C values of 0.6 (Table 5) is very similar to the average of the observed C values of 0.59 (Table 4).

Line 201-214 The correlation of the simulated and the observed BFI values respectively with forest cover showed different slope (Fig. 5a and 6). As an example, to achieve a BFI value of 0.5, the required proportion of forest cover based on the simulated BFI was

45 % (Fig. 5a). Meanwhile, to achieve a similar BFI values, the required proportion of forest cover based on the observed values was 33% (Fig. 6). Thus, the SWAT model underestimated the simulated BFI value. This can be explained by the fact that the SWAT model (version 2012) considered only shallow groundwater in the streamflow simulation (Neitsch et al., 2009). The observed BFI on the other hand included deep groundwater flow as well.

————————————————————————————————-

28) Lines 168-173: More suitable in the methodology section.

We agree with the referee suggestion. We have moved the sentences to the methods section (Subsection 2.2.1)

————————————————————————————————-

29) Lines 180-184 Too long sentence, it is better to follow simple sentences. Improve the language as well.

Revision made. We have improved the language.

Line 180-184 Infiltration data in different land-use types from the study area (Tarigan et al. 2016) showed the following order: oil palm harvest path (3 cm h-1) < rubber (7 cm h-1-7.8 cm h-1) < forest (47 cm h-1). Low infiltration capacity in oil palm and rubber plantations was the reason for higher C values in the sub-watersheds with high proportions of the plantation land use.

————————————————————————————————-

30) Line 182: Oil palm harvest and oil palm circle are equal (i.e. 3 cm h-1).

Revision made. We have improved the language

————————————————————————————————-

31) Lines 185-188: I'm puzzled by this conclusion. Is the rainfall distribution similar

throughout the basin? Because if there is a spatial variation in rainfall magnitude, the effects of forest conversion on the flow regulation would vary accordingly.

We thank the referee for raising this question. We agree that there is always spatial variation in rainfall magnitude throughout the basin from one event to another event. The SWAT model is considered as long-term yield model and not an event-based model. In addition, both watershed in our study were partitioned into 48 sub-watershed, which reduce the degree of rainfall spatial variability in the watersheds.

————————————————————————————————————-

32) In Figure 4a, I see a C value less than 0.35 for forest cover about 20%, what do you think about this?

The mentioned sub-watershed in Figure 4a is the sub-watershed with number 19 in the BH watershed. The reason why the C value is low despite the low forest cover was due to the fact that the proportion of oil palm and rubber plantation in this sub-watershed is 0 %. In addition, the proportion of agroforest in this sub-watershed is 30% which help reducing the C value.

————————————————————————————————————-

33) Line 207: please improve the language

Revision made. We have improved the language. The correlation of the simulated and the observed BFI values respectively with forest cover showed different slope (Fig. 5a and 6).

————————————————————————————————————-

34) Lines 207-214. I think this need more discussion. SWAT has a known limitations in simulating the low flow regime and that would have an effect on the BFI, as also mentioned by the authors. See the recent study for further discussion: Pfanner-still, M., B. Guse, and N. Fohrer, 2014a. A Multi-Storage Groundwater Concept for

the SWAT Model to Emphasize Non- linear Groundwater Dynamics in Lowland Catchments. Hydro- logical Processes 28:5599-5612, DOI: 10.1002/hyp.10062

We agree with the referee, the SWAT version used in this study has limitation to model the groundwater component of the streamflow, especially the watershed with significant proportion of groundwater in the total flow. We have enriched the discussion with the suggested literature. (see the description in the following paragraphs)

Line 211: The SWAT model underestimated the simulated BFI value in our study area. The reason is that the SWAT model considered only shallow groundwater in the stream flow simulation (Neitsch et al., 2009). Meanwhile, the observed BFI included deep groundwater flow. To improve the performance of the SWAT model for deep groundwater flow (low flow) simulation, Pfannerstill et a.l, (2014) modified groundwater module by splitting the active groundwater storage into a fast and a slow contributing aquifer. The result of this modification leads to better prediction of low flow. Bailey et al. (2017) coupled SWAT with physically-based, spatially-distributed groundwater model (MODFLOW) to improve groundwater flow process in SWAT.
* * *
35) Line 344: ": : :MT(b,: : :"

Revision made. We thank you the referee for the correction
* * *
36) Line 376: Table 3, In MT watershed sub.wat.nr 23 has a 100% forest cover but the BFI is low, meaning low baseflow contribution from the groundwater. Justify this in the discussion.

The sub-watershed nr. 23 is the only sub-watershed in MT watershed with high proportion of steeper slope (76% of the sub-watershed). The steep slope increased the C value and decrease the BFI. Among 48 considered sub-watersheds, only 2 of them have such a kind of slope characteristics. This type of sub-watersheds is normally situated at the upper-mountainous catchment area. These area are not suitable for the oil palm development.
* * *
37) Line 379 Table 4, Please recheck the numbers and the calculations.

We have re-checked and corrected the errors. The total average of the C values of 0.59 remains unaffected.
* * ** * *
[Figure]

[Figure]

Figure 1. Soil type in BH and MT watersheds. Soil types respresent Fluvaquents (S-3), Humitropepts (S-4), Paleudults (S-7), Tropofluvents (S-10), Troposaprist (S-11), Tropodults (S-12), Dystrandepts (S-13), Dystropepts (S-14).

**Fig. 1.**

Figure 2. Oil palm hydrological components

**Fig. 2.**

[Figure]

Figure 3. Interception of different plantation crops and forest. Different letters indicate significant differences of averages according to a Bonferroni-corrected posthoc t-test based on an ANOVA ($p< 0.05$)

**Fig. 3.**

[Figure]

Fig. 4. Surface runoff of different land use types in the study area. The different letters indicate significant differences among averages according to a Bonferroni-corrected posthoc t-test based on an ANOVA (p< 0.05).

**Fig. 4.**

[Figure]

[Figure]

Figure 5. Observed vs. simulated streamflow and 95% uncertainty interval (95PPU) of behavioral simulations over time (defined as simulations with Nash-Sutcliff efficiency NSE > 0.5) for the BH (a) and MT (b) watersheds respectively.

**Fig. 5.**

Table 1. Model input data sources for the watershed modeling

| Data type | Resolution | Description | Source Data |
|---|---|---|---|
| Topography | 30 m | Digital Elevation Model with 30 m pixel resolution | LAPAN |
| Soils | 1:250,000 | Soil hydraulic conductivity, bulk density, available water content and texture were resampled in the field | Soil Research Institute, Ministry of Agriculture |
| Land use | 1:100,000 | Land use map with intensive ground check | Regional Planning office (BAPPEDA) |
| Rainfall and climate | Daily | Rainfall stations (Rantau Pandan, Siulak Deras, Muara Imat); climate station (Jambi, Pematang Kabau and Bungku) | BMG office (Meteorology and Geophysics Agency) and CRC990 |
| Streamflow | Daily discharge data | Stations: Muara Tembesi, Rantau Pandan, Air Gemuruh, Batang Tabir, Batang Pelepat, Muara Kilis | Ministry for Public work (BBWS) |

LAPAN: National Aeronautics and Space Agency (*Lembaga Antarikasa dan Penerbangan Nasional*), BAPPEDA: Regional Planning Agency (*Badan Perencanaan Daerah*), BMG: Meteorology and Geophysics Agency (*Badan Meteorologi dan Geofisika*), CRC990: Collaborative Research Centre 990, BBWS: Catchment Regional Agency (*Balai Besar Wilayah Sungai*)

**Fig. 6.**

Table 2. Adapted input of crop-parameters different land-use types

| SWAT input parameters | | Oil palm | Rubber | Agro-forest | Forest |
|---|---|---|---|---|---|
| CANMX | Original | 0 | 0 | 0 | 0 |
| | Adapted | 4.0 | 2.7 | 4.3 | 5.8 |
| OV_N* | Original | 0.14 | 0.14 | 0.1 | 0.1 |
| | Adapted | 0.07 | 0.14 | 0.4 | 0.5 |
| BLAI | Original | 5 | 2.6 | 5 | 5 |
| | Adapted | 3.6** | - | - | - |
| CHTMX | Original | 3.5 | 3.5 | 6 | 10 |
| | Adapted | 12 | | | |
| T_BASE | Original | 7 | 7 | 0 | 10 |
| | Adapted | 20 | 20 | 20 | 20 |
| T_OPT | Original | 20 | 20 | 30 | 30 |
| | Adapted | 28 | 28 | 30 | 30 |

\* OV_N values are low in oil palm and rubber due to the clean weeded management practice
\*\* Meijide et al. (2017)

**Fig. 7.**

Table 3. Adapted CN values for typical land use types in the study area

| Landuse | | Hydrologic Soil Group | | | |
|---|---|---|---|---|---|
| | | A | B | C | D |
| Oil palm | Original CN value | 45 | 66 | 77 | 83 |
| | Adapted CN value | - | - | - | **83** |
| Rubber | Original CN value | 45 | 66 | 77 | 83 |
| | Adapted CN value | - | - | - | **83** |
| Agroforest (FRST) | Original CN value | 36 | 60 | 65 | 79 |
| Forest | Original CN value | 25 | 45 | 70 | 77 |

**Fig. 8.**

Table 4. The calibration parameters used in the study, including the initial value range and the final calibrated values for the BH and the MT watersheds respectively.

| Parameters | Descriptions | Initial value range | Best fit values | |
|---|---|---|---|---|
| | | BH and MT | BH | MT |
| ALPHA_BF | Baseflow recession constant | 0.0 – 1.0 | 0.94 | 0.91 |
| GW_DELAY | Groundwater delay time (days) | 30 - 450 | 62.5 | 57.2 |
| GWQMN | Water depth in a shallow aquifer for a return flow (mm H2O) | 0.0 – 2.0 | 0.99 | 0.45 |
| GW_REVAP | Evaporation from the ground water (mm) | 0.0 - 0.02 | 0.13 | 0.07 |
| CH_N2 | Manning's "n" value for the main channel | 0.0 – 0.3 | 0.05 | 0.15 |
| CH_K2 | Eff. hydraulic conductivity in the main channel alluvium (mm/hr) | 5.0 - 130 | 35.6 | 24.4 |
| SOL_AWC | Available water capacity of the soil (mm H2O/mm soil) | - 0.2 – 0.4 (V)[a] | 0.09 | 0.04 |
| SOL_K | Saturated hydraulic conductivity (mm h[-1]) | - 0.8 – 0.8 (V)[a] | 0.71 | 0.12 |
| OV_N | Manning's "n" value for overland flow | - 0.2 – 1.0 (V)[a] | 0.51 | 0.29 |

[a] (V) = Variable depending on land-use and soil, changes in calibration were therefore expressed as fraction

**Fig. 9.**

---

## Author Comment (AC10) · 22 Aug 2017

General comments: I found the topic and results described in this manuscript to be quite interesting. There is very limited information available in the literature to date regarding the potential effects of expanded production of rubber or oil palm trees, using SWAT model or any other modeling approach. Thus I think that the information reported in this manuscript will ultimately prove to be a useful contribution to Hydrology and Earth System Sciences (HESS) and the existing literature in general. However, I believe that the current manuscript suffers from several deficiencies including

inadequate review of existing literature, insufficient description of SWAT and key input parameters (including coefficients used for rubber tree and oil palm tree in the crop parameter file), lack of in-depth description of SWAT calibration and validation results, and an inadequate description of the simulated watersheds. Specific comments regarding these issues are provided below.

We appreciate the referee's concerns. We have addressed all referees' concern in the respective comments below including: a) more comprehensive review of existing literature, b) in depth description of key crop parameters. In addition, we have restructured method section into several subsections as follows. Each subsection is described in the respective specific comment below.

2. Methods

2.1 Study area

2.1.1. Land use and soil characteristics

2.1.2 Watershed characteristics

1. Macro watersheds

2. Small watersheds

2.2 Flow simulation

2.2.1 SWAT model

1. Crop and Soil parameters

2. Input data

3. Model validation and calibration

2.2.2 Simulated C and BFI values

2.2.3 Observed C and BFI values
1. Observed C values

2. Observed BFI values
* * *
Specific comments

1) Abstract: The Abstract needs to be considerably revised to reflect more of the actual quantitative results of the study versus the "general discussion" that dominates much of the abstract between lines 9 to 24. The revised abstract should include a summary of the baseline calibration and validation results.

We agree with the referee's suggestions, and have improved the abstract accordingly. (see the description in the following paragraphs)

Lines 9-24: In many tropical regions, rapid expansion of monoculture plantations has led to a sharp decline of forest cover potentially degraded the water flow regulation function of watersheds. In a watershed where expansion of agricultural plantations occurs rapidly, the regional planner need to know the minimum proportion of forest cover required to maintain proper water flow regulation function of a watershed. Research dealing with this issue is still rare, especially in the tropical area where oil palm expansion occurs at alarming rate. We investigated the impact of forest and monoculture plantations (oil palm and rubber plantations) on rainfall partitioning to direct runoff and subsurface flow for a humid tropical watershed in Indonesia. The results are based on streamflow as simulated by a calibrated SWAT model and observations across several watersheds and subsequently derived the direct runoff coefficient (C) and the baseflow index (BFI). The model gave satisfactory performance with the NSE values of 0.80-0.88 (baseline calibration) and 0.80 - 0.85 (validation); and the PBIAS values of -2.9 - 1.2 (calibration) and 7.0-11.9 (validation). The study exhibits a statistically significant correlation of percentage of forest covers in a watershed with C (negatively) and BFI (positively). On the other hand, the rubber and oil palm plantations showed flow regulation behavior contrary to forest covers. Finally the study suggests the minimum forest cover requirement in the study area (i.e. 30%) for sustainable ecosystem services. The new contribution of our study is the establishment of quantitative relation between forest cover and flow indicators in a watershed, which can be used as a guide for regional planners to determine the minimum proportion of forest conservation area to maintain a sustainable ecosystem service of water flow regulation in a watershed.

————————————————————————

2) Lines 43-45: I would suggest you rewrite this sentence to read something like: "This vertical movement of water in the soil determines how much water flows as direct runoff and how much percolates to the water table where it sustains baseflow or groundwater (references)."

We agree with the referee's suggestions, and have revised the sentence.

————————————————————————

3) Lines 49-68: Please include citation and discussion of some "big picture" studies regarding the impacts of Palm Oil and/or Rubber Trees in the southeast Asia region such as those listed immediately below.

We thank the referee for this suggestion. We have substantially improved the citation and discussion of some studies regarding the impacts of palm oil and/or rubber trees in the Southeast Asia region.

Line 30-46 In Southeast Asia in particular, the land under oil palm and rubber have expanded considerably. In Indonesia, which is now the largest palm oil producer worldwide, the oil palm area increased from 0.7 million ha in 1990 to 11 million ha in 2015 (Ditjenbun, 2013; Tarigan et al., 2016). Projections of additional land demand for palm oil production in 2020-2050 ranges from 1 to 28 Mha in Indonesia (Wicke et al., 201, Afriyanti et al., 2016). The rapid increased of oil palm expansion is partly triggered by increased demand for biofuel production (Mukherjee and Sovacoo. 2014). While

oil palm has improved farmer and regional economic, it has been subject to the environmental and social criticism. Oil palm expansion is often held responsible for deforestation (Wicke et al., 2011; Gatto et al., 2015; Vijay et al., 2016), biodiversity loss (Koh and Wilcove, 2008; Fitzherbert et al., 2008;Wilcove and Koh, 2010; Carlson et al., 2012; Krashevka et al., 2015), decreased soil carbon stock (Guillaume et al., 2015, 2016; Pransiska et al., 2016) and increased greenhouse gas emissions (Allen et al., 2015; Hassler et al., 2017). Apart from oil palm, another prevalent plantation crop in Southeast Asia is rubber plantation (Ziegler, et al., 2009). In Indonesia itself, the rubber plantation covers 3.5 million hectares of land (Ditjenbun, 2013). The land devoted for rubber in Southeast Asia could double or triple by the year 2050 (Ziegler, et al., 2009). Expansion of the rubber plantation reduces soil infiltration capacity, accelerates soil erosion, increases stream sediment load (Ziegler, et al., 2009; Tarigan, et al., 2016), increased evapotranspiration (Wangpimool et al., 2017) and decreases soil carbon stock (Ziegler et al., 2011).

Some newly added References:

Afriyanti, D., Kroeze, C., Saad A.2016. Indonesia palm oil production without deforestation and peat conversionby 2050. Science of the Total Environment 557–558 (2016) 562–570

Allen K, Corre MD, Tjoa A, Veldkamp E (2015) Soil nitrogen-cycling responses to conversion of lowland forests to oil palm and rubber plantations in Sumatra, Indonesia. PLoS ONE 10(7): e0133325. http://journals.plos.org/plosone/article?id=10.1371/journal.pone.0133325

Carlson, K.M., Curran, L.M., Ratnasari, D., Pittman, A.M., Soares, B.S., Asner, G.P., Trigg, S.N., Gaveau, D.A., Lawrence, D., Rodrigues, H.O.. 2012. Committed carbon emissions, deforestation, and community land conversion from oil palm plantation expansion in West Kalimantan, Indonesia. Proc Nat Acad Sci USA 109:7559–7564.

Fitzherbert, E. B., Struebig, M. J., Morel, A., Danielsen, F., BruÌLhl, C. A., Donald, P. F.,

Phalan, B., 2008. How will oil palm expansion affect biodiversity? Trends in Ecology & Evolution 23 (10), 538–545.

Gatto M, Wollni M, Qaim M (2015) Oil palm boom and land-use dynamics in Indonesia: The role of policies and socioeconomic factors. Land Use Policy 46: 292-303. http://www.sciencedirect.com/science/article/pii/S0264837715000733

Guillaume T, Damris M, Kuzyakov Y (2015) Losses of soil carbon by converting tropical forest to plantations: Erosion and decomposition estimated by $\delta$ 13 C. Global Change Biology 21: 3548-3560. http://onlinelibrary.wiley.com/doi/10.1111/gcb.12907/abstract

Guillaume T, Holtkamp AM, Damris M, Brümmer B, Kuzyakov Y (2016) Soil degradation in oil palm and rubber plantations under land resource scarcity. Agriculture, Ecosystems & Environment 232: 110-118 http://www.sciencedirect.com/science/article/pii/S0167880916303619

Hassler E, Corre MD, Tjoa A, Damris M, Utami SR, Veldkamp E (2015) Soil fertility controls soil-atmosphere carbon dioxide and methane fluxes in a tropical landscape converted from lowland forest to rubber and oil palm plantations. Biogeosciences 12: 5831-5852. http://www.biogeosciences.net/12/5831/2015/bg-12-5831-2015.html

Koh, L.P., Wilcove, DS., 2008. Is oil palm agriculture really destroying tropical biodiversity? Conservation Letters 1 (2008) 60–64

Krashevska V, Klarner B, Widyastuti R, Maraun M, Scheu S (2015) Impact of tropical lowland rainforest conversion into rubber and oil palm plantations on soil microbial communities. Biology and Fertility of Soils 51: 697-705. http://link.springer.com/article/10.1007/s00374-015-1021-4

Mukherjeea, I., B.K. Sovacoo. 2014. Palm oil-based biofuels and sustainability in southeast Asia: A review of Indonesia, Malaysia, and Thailand. 37: 1-12. DOI: 10.1016/j.rser.2014.05.001.

Pransiska Y, Triadiati T, Tjitrosoedirjo S, Hertel D, Kotowska MM (2016) Forest conversion impacts on the fine and coarse root system, and soil organic matter in tropical lowlands of Sumatera (Indonesia). Forest Ecology and Management 379: 288-298 http://www.sciencedirect.com/science/article/pii/S0378112716303942

Vijay V, Pimm SL, Jenkins CN, Smith SJ. 2016. The Impacts of Oil Palm on Recent Deforestation and Biodiversity Loss. PLoS ONE 11 (7): e0159668. doi:10.1371/journal.pone.0159668

Wicke B, Sikkema R, Dornburg V, Faaij A. 2011. Exploring land use changes and the role of palm oil production in Indonesia and Malaysia. Land Use Policy 28 (2011):193–206

Wilcove DS, Koh LP. 2010 Addressing the threats tobiodiversity from oil-palm agriculture. Biodivers.Conserv. 19, 999–1007. (doi:10.1007/s10531-009-9760-x)

Wilcove et al. 2013. Navjot's nightmare revisited: logging, agriculture, and biodiversityin Southeast Asia. Trends in Ecology and Evolution 28(9): 531-540. DOI:10.1016/j.tree.2013.04.005.

Ziegler et al. 2009. The Rubber Juggernaut. Science 324: 1024–1025. DOI: 10.1126/science.1173833.

Ziegler et al. 2011. Recognizing Contemporary Roles of Swidden Agriculture in Transforming Landscapes of Southeast Asia. Conservation Biology 25(4): 846-848. Available at: http://www.jstor.org/stable/27976544.

————————————————————————

4) Lines 69-70: Please expand this discussion to provide a broader review of different modeling and other analysis methods, beyond the option of SWAT, available to assess the impacts of expanded rubber and oil palm plantations in the Southeast Asia region.

We thank the referee for this suggestion, and have expanded the discussion to provide a broader review of different modeling and other analysis methods, beyond the option

of SWAT.

Lines 78-92: A number of approaches can be used to assess the impacts of expanded rubber and oil palm plantations on hydrological characteristics in the Southeast Asia region. The approaches can be categorized as empirically-based and process-based. Empirical-based approaches use long-term historical data to correlate the land use changes with the corresponding streamflow data (Adnan and Atkinson, 2010; Rientjes et al., 2011; Mwangi et al. 2016) or paired catchment studies (Bosch and Hewlett, 1982; Brown et al., 2005). Process-based method utilizes physically based hydrological models where the change impact is determined by varying land use/cover settings (Guo et al. 2016; Khoi et al. 2014; Zhang et al. 2016, Marhaento et al. 2017, Wangpimool et al. 2017). Process-based approach require more data as input and subject to high uncertainty in parameter estimation (Zhang et al. 2016; Xu et al. 2014). Due to the absence of long-term historical data in our study area, we used the second approach. The distributed hydrologic models such as the Soil and Water Assessment Tool (SWAT) eco-hydrological model (Arnold et al., 1998; 2012) are useful to understand the effects of land use changes on watershed flow regulation. It quantifies the water balance of a watershed on a daily basis (Neitsch et al., 2009) and has been recommended for the evaluation of hydrological ecosystem services of a watershed (Vigerstol et al., 2011). The SWAT modeling approach is one of the most widely used and scientifically accepted tools to assess the water management in a watershed (Gassman et al., 2007).

Some newly added References:

Adnan, N. A., Atkinson, P. M. 2011. Exploring the impact of climate and land use changes on streamflow trends in a monsoon catchment. International Journal of Climatology 31, 815–831.

Guo J, Su X, Singh VP, Jin J. 2016. Impacts of Climate and Land Use/Cover Change on Streamflow Using SWAT and a Separation Method for the Xiying River Basin in Northwestern China. Water 2016, 8, 192; doi:10.3390/w8050192.

Khoi DN, Suetsugi T. 2014. Impact of climate and land-use changes on hydrological processes andsediment yield Ťa case study of the Be River catchment, Vietnam. Hydrological Sciences Journal–Journal des Sciences Hydrologiques, 59 (5) 2014.

Marhaento et al. 2017. Attribution of changes in the water balance of a tropical catchment to land use change using the SWAT model. Hydrological Processes. 31(11):2029–2040. DOI: 10.1002/hyp.11167.

Meijide A, Röll A, Fan Y, Herbst M, Niu F, Tiedemann F, June T, Rauf A, Hölscher D, Knohl A (2017) Controls of water and energy fluxes in oil palm plantations: Environmental variables and oil palm age. Agricultural and Forest Meteorology 239: 71-85 http://www.sciencedirect.com/science/article/pii/S0168192317300771

Rientjes, T. H. M., Haile, A. T., Kebede, E., Mannaerts, C. M. M., Habib, E., & Steenhuis, T. S. (2011). Changes in land cover, rainfall and stream flow in Upper Gilgel Abbay catchment, Blue Nile basin–Ethiopia. Hydrology and Earth System Sciences, 15(6), 1979–1989.

Wangpimool et al. 2017. The impact of Para rubber expansion on streamflow and other water balance components of the Nam Loei River Basin, Thailand. Water. 9(1) DOI: 10.3390/w9010001.

Xu X, Yang D, Yang H, Lei H. 2014. Attribution analysis based on the Budyko hypothesis for detecting the dominant cause of runoff decline in Haihe basin. Journal of Hydrology 510 (2014) 530–540.

Zhang L, Nan Z Xu Yi, Li S. 2016. Hydrological Impacts of Land Use Change and Climate Variability in the Headwater Region of the Heihe River Basin, Northwest China. PLoS One. 2016; 11(6):e0158394. doi: 10.1371/journal.pone.0158394.
* * *
5) The expanded paragraph noted in comment 3 should be followed by a specific paragraph about SWAT including relevant review studies about SWAT and a more in-depth

review of how SWAT has been used for land use change analyses. Note that the Zhang et al. (2013) article you cite in line 76 is not a very good choice regarding reviews of SWAT studies; please instead cite one or more of the studies listed on the webpage at http://swat.tamu.edu/publications/special-issues/ or in the "SWAT Publications box" in http://swat.tamu.edu/. Please also cite some relevant SWAT "land use change studies" (see the SWAT Literature Database that can again be accessed on the SWAT model homepage) such as those listed here:

Babel, M.S., B. Shrestha and S.R. Perret. 2011. Hydrological impact of biofuel production: A case study of the Khlong Phlo Watershed in Thailand. Agricultural Water Management. 101(1): 8-26. DOI: 10.1016/j.agwat.2011.08.019. Marhaento et al. 2017. Attribution of changes in the water balance of a tropical catchment to land use change using the SWAT model. Hydrological Processes. 31(11):2029–2040. DOI: 10.1002/hyp.11167. Tan et al. 2015. Impacts of land-use and climate variability on hydrological components in the Johor River basin, Malaysia. Hydrological Sciences Journal. 60(5): 873-889. DOI: 10.1080/02626667.2014.967246. Tarigan et al. 2016. Mitigation options for improving the ecosystem function of water flow regulation in a watershed with rapid expansion of oil palm plantations. Sustainability of Water Quality and Ecology . 8: 4-13. DOI: 10.1016/j.swaqe.2016.05.001. Wangpimool et al. 2017. The impact of Para rubber expansion on streamflow and other water balance components of the Nam Loei River Basin, Thailand. Water. 9(1) DOI: 10.3390/w9010001.

We thank you the referee for suggestions. We have added specific paragraph about SWAT and a more in-depth review of how SWAT has been used for land use change analyses.

Line 93-102 Marhaento et al. (2017) used the SWAT model to simulate impact of forest cover and agriculture land use on the runoff coefficient and the ratio of base flow to stream flow in Java Island Indonesia and found that forest cover change from 48.7% to 16.9% resulted in the increased of the runoff coefficient (C) to 44.6% and decrease of the ratio of base flow to stream flow to 31.1% showing similar trend with that of our

results. Meanwhile, Wangpimool et al., (2017) found that annual reduction of the basin average water yield about 3% based on the SWAT model simulation due to the rubber expansion in Thailand from 2002 to 2009. Babel et al., (2011) simulated impact of oil palm expansion using SWAT in Thailand and reported increased nitrate loading (1.3 to 51.7%) to the surface water. The new contribution of our study is the establishment of quantitative relation between forest cover and flow indicators in a watershed, which can be used as a guide for the regional planners to determine the minimum proportion of forest conservation area to maintain a sustainable ecosystem service of water flow regulation in a watershed. Tarigan et al. (2016) used SWAT model to simulate impact of soil and water conservation practices on low flow in oil palm dominated watersheds in Jambi Provinces, Indonesia.

———————————————————————————

6) Lines 71-73: These two current sentences have grammatical problems. As a part of comment 4, I suggest that you revise the text as follows: "A useful tool to answer this question is the Soil and Water Assessment Tool (SWAT) ecohydrological model (Arnold et al., 1998; 2012), which quantifies the water balance of a watershed on a daily basis (Neitsch et al., 2009) and has been recommended for the evaluation of hydrological ecosystem services of a watershed (Vigerstol et al., 2011)."

We agree with the referee's suggestions, and have improved the sentences accordingly.

Lines 71-73 Distributed hydrologic models such as the Soil and Water Assessment Tool (SWAT) eco-hydrological model (Arnold et al., 1998; 2012) are useful to understand the effects of land use changes on watershed flow regulation. It quantifies the water balance of a watershed on a daily basis (Neitsch et al., 2009) and has been recommended for the evaluation of hydrological ecosystem services of a watershed (Vigerstol et al., 2011).

———————————————————————————

7) Study area description: The two study watersheds should be described in depth in this subsection rather than being referenced later in subsection 2.2 (please describe the area of the watersheds in km2 rather than ha). More detailed land use information (percentages of each type of land use) for the two watersheds should be provided (rather than waiting until subsections 2.3.1 and 3.2 to describe some of that information), as well as more information about the natural vegetation, and rubber and oil palm plantations (growth cycles, management practices, time period of plantation development, etc.). Further details about the typical porosity and other characteristics of the soils in the study watersheds would also be useful.

We thank the referee for the suggestion. We have re-restructured and substantially improved the description of the whole subsections in the method section. We describe subsection 2.1.1:1 (Land use and soil characteristics) in this comment (nr. 7); subsection 2.1.2:1 (Watershed characteristics) in comment nr. 8; subsection 2.1.2:2 (Small watersheds) in comment nr. 9.

2. Methods

2.1 Study area

2.1.1. Land use and soil characteristics

2.1.2 Watershed characteristics

2.1 Study area The study area was situated in the Jambi Province of Sumatra with geographic location of 1o54'31.4"S, 103o16'7.9" E covering area of 31,868 sq km. The area is experiencing rapid development of plantations, mainly oil palm and rubber plantations (Drescher et al., 2016). The climate is tropical humid with average temperature of 27 °C and average rainfall of 2700 mm yr-1. Rainy season occurs during October until March. Flooding events occur normally in the months of January and February. A dry season with monthly precipitation less than 100 mm occurs from June until September. The soil types in the study area are dominated by clay Acrisols (Allen et al., 2015).

2.1.1. Land use and soil characteristics. Based on the land use map (Bappeda 2013; Ditjenbun 2013), the dominant land uses in BH and MT watersheds respectively are forest 50%, 30%; oil palm; 4%, 32% ; rubber 14%, 10%; (rubber) agroforest 6%, 9% and shrubland 11%, 10%. Oil palm and rubber are the most important plantation crops in this area. Both crops are planted in rows with planting distance 8 m and 4 m respectively. The weeds in the plantation were regularly eradicated with herbicides. Pruning, i.e. cutting of oil palm fronds, is common practice in oil palm cultivation. The pruned leaf fronds are stacked in the middle of the row between two trees forming frond piles. The width of frond piles is normally ∼2 m. Based on our field measurement in the study area, oil palm and rubber plantation soils showed significantly higher bulk densities than soils in forest (Figure 1). Higher bulk density in monoculture plantation can be explained by compaction due to frequent harvest activities taking place at least 2 times per month under oil palm and several days a week in rubber plantations. Similar to our findings, Sunarti et al. (2008) found that bulk density in forest soil (0.81 gr cm-3) was significantly lower than in oil palm (1.05 gr cm-3) and rubber plantation (1.14 gr cm-3) in Bungo District, Jambi. Tanaka et al. (2008) also found higher soil bulk density in oil palm plantations compared to secondary forest in Sarawak, Malaysia.
* * *
8) In relation to comment 7, some description of all six macro watersheds shown in Figure 1 should also be provided in the Study area description subsection. Who defined these six watersheds and why? It is clear that hydrologic data was collected for the watersheds but the current text is vague regarding the overall purpose of these six watersheds.

We agree with the referee's suggestions, and have described all six macro watersheds under subsection 2.1.2 (Macro watershed).

2. Methods

2.1 Study area

2.1.1. Land use and soil characteristics

2.1.2 Watershed characteristics

1. Macro watersheds

2. Small watersheds

2.1.2 Watershed characteristics The study watershed consists of 2 categories (Figure 2): 1) two macro watershed (BH and MT) used for flow simulation with SWAT model, and 2) six macro watershed (including BH and MT) for BFI field data observation. The area of BH and MT watersheds were 18,415 km2 and 13,452 km2, respectively. Both watersheds were chosen as representative of the rapid land use transformation from forest to plantation in Indonesia. Beside BH and MT watersheds used for SWAT modeling, we also collected time series streamflow data from four nearby watersheds. The streamflow data from these six watersheds (including from BH and MT watersheds) were used to determine observed BFI values and then correlated with forest cover proportion in the respective watersheds. This correlation was compared with that obtained from the SWAT simulation model to get qualitative impression whether the BFI values calculated form the SWAT model really reflects the field observation (despite good performance of the model in our study).

————————————————————————

9) Also in relation to comment 7, please describe the "small watersheds" referenced in lines 144-145 and 195-196 and shown in Figure 1 in the study area subsection, rather than waiting to describe those in current section 2.3.1 (and that information does not need to be repeated at the start of section 3.2). What other hydrologic data were collected for those small watersheds besides the C values?

We agree with the referee's suggestions, and have described "small watersheds" in subsection 2.1.2 (Small watershed).

Parallel to the purpose of collecting observed BFI values from the six-macro watershed

in the comment 8, we also collected data from two small watershed in the study area to determine observed C values. These values were compared with that obtained from the SWAT simulation model to get qualitative impression whether C values calculated form the SWAT model really reflects field observation (despite good NSE performance of the model).

The dominant land-use type in the first small watershed was oil palm (90%), meanwhile 80% of the second watershed was covered by rubber plantations and the rest by rubber agroforest. Both watersheds were instrumented with rectangular weirs and automatic water level recorders. The direct runoff components of the hydrographs were separated by using the straight line method described in (Blume et al., 2007). After hydrograph separation, we calculated the direct runoff coefficient (C). The direct runoff coefficient C is the percentage of rainfall that appears as surface runoff during a rainfall event, or directly following a rainfall event. We did not calculate BFI values along with C values in the small watershed experiments, because BFI calculation requires longer hydrograph records. The hydrograph records of the small watersheds were available in the time period 2013-2015.

––––––––––––––––––––––––––––––––––––––––––

10) Please rewrite "C&BFI" as "C and BFI" throughout the text.

Revision made

––––––––––––––––––––––––––––––––––––––––––

11) A SWAT Description subsection needs to be added to the manuscript. This should note the specific version of the model used for the study (including the Revision number) and provide a succinct overview of the model, especially regarding components that were particularly important for the study you conducted. A description of the crop parameters used for the rubber and oil palm trees, and other vegetation in the watersheds, should also be provided (those parameters could be described later in the

methods if more appropriate). See the Wangpimool et al. article listed in comment 4 above regarding revised rubber tree crop parameters they used in their study.

We agree with the referee's suggestions, and have described the SWAT model and crop parameter in subsection 2.2.1.

2.2.1 SWAT model

We used the SWAT model version 2012 (Arnold et al., 2012). The SWAT model is a continuous model, i.e. a long-term yield model. The model was developed to simulate the impact of land cover/management practices on the streamflow in complex watersheds with varying soil, land use and management condition over long periods of time. Major model components include weather, hydrology, soil temperature and properties, plant growth, nutrients, and land management (Arnold et al., 2012; Neitsch et al., 2009). Delineation ofwatersheds and their sub-watersheds in our study area was carried out automatically by the SWAT model and was based on a digital elevation model (DEM) with a 30-m resolution. During the automatic delineation we pre-defined an area of 50.000 ha as a threshold for a minimum sub-watershed area. Based on this threshold, both study watersheds in our study area were further sub-divided into 25 and 23 sub-watersheds, respectively. The sub-watershed is further sub-divided into hydrological response units (HRUs) with homogeneous hydrological unit defined by topography, soil, and land use characteristics. Hydrological outputs are then calculated in the HRUs based on the water balance equation. Output of the SWAT model include total stream flow, surface flow and base flow. These output were used to calculate the C and BFI values for each sub-watershed. For this simulation, the SWAT model required other inputs such as climate data, as well as soil and land-use maps for each sub-watershed (Table 1).

We also carried out field data collection on several important parameters including hydraulic conductivity (SOL_K), bulk density (SOL_BD), available water content (SOL_AWC) and texture for SWAT model input.

Digital Elevation Model with 30 m pixel resolution is available from the National Aeronautics and Space Agency. Rainfall and climate date are available from the Meteorology and Geophysics Agency. The streamflow data of the six macro watersheds were provided by the Ministry for Public work. The land use data are available from the Regional Planning office. All these data are freely available for research purposes by official request to the corresponding institutions. The time series streamflow and the rainfall records for the small catchments, the resampled soil hydraulic conductivity, bulk density, available water content and texture are deposited by the first author office at Bogor Agricultural University and EFForTS Database (https://efforts-is.uni-goettingen.de). The land-use and soil map for the study area was obtained from Jambi Province Regional Planning (BAPEDA, 2013) and Agricultural Plantation offices (Ditjenbun, 2013). Soils in our study area are dominated by two soil types, namely Tropodult and Dystropept (Figure 3).

Crop parameter

According to Griensven et al. (2014), the SWAT is designed for temperate regions so that it is necessary to adapt the crop parameters for application in a tropical region. In this respect, we adjusted the crop parameters, directly related to the flow component such as CANMX and CN. To adapt these values we carried out field measurement on several important hydrological component including interception, infiltration, and overland flow (Figure 4).

Canopy Storage (CANMX)

Interception reduces the amount of water reaching the ground and consequently reduces streamflow. We measured interception in oil palm, rubber, agroforest, and forest trees at the plot between November 2012 and February 2013. In total there were 30 rainfall events during this time, representing light to heavy rain. In oil palm, rainfall is not only intercepted by leaves and branches but also by hollow spaces between fronds and trunk. This type of interception is called trunk storage and may have led to the

slightly increased interception in oil palm. Interception in oil palm was rather similar to interception in the forest (Figure 5).

The measured interception values were used as an estimate of (CANMX), which serves as an input parameter for the SWAT model. The CANMX is the maximum amount of water that can be trapped in the canopy and trunks when they are fully developed. Higher CANMX values reduce potential runoff during heavy rains. Beside CANMX we also adapted other crop parameters such as OV_N, BLAI. CHTMX, T_BASE and T_OPT (Table 2).

Adapted CN values for oil palm and rubber land uses.

One important parameter of the SWAT model related to surface runoff modeling is SCS curve number (CN, Arnold et al., 2012). It determines proportion of rainfall becoming surface runoff. Its value range from 0-100. The bigger the value the higher the proportion of surface run of on a particular rainfall event. The SCS curve number (CN) is differentiated into Hydrologic Soil Groups (HSG) A,B,C, and D which are a function soil's infiltration. We measured soil infiltration and surface runoff in the typical land use types in our study area i.e. oil palm, rubber, and forest. Infiltration was measured using a double-ring infiltrometer. No infiltration measurement was carried out under agroforest as infiltration in agroforest is likely similar to infiltration in secondary forest. Infiltration measurements in different land-use types from the study area showed the following order: oil palm harvest path (3 cm h-1) < rubber (7-7.8 cm h-1) < forest (47 cm h-1). The infiltration in the oil palm, rubber plantations were markedly lower than those at the forest. The surface runoff in oil palm and rubber plantation were significantly higher than those in agroforest and forest (Figure 6). Low infiltration capacity in oil palm and rubber plantations was one reason for higher surface the plantation land use (Tarigan et al. 2016).

Due to the high surface runoff and the infiltration rate, we adopted HSG-D category for all HRUs in oil palm and rubber land uses irrespective of soil types (Table 3). For forest

and agroforest, we assumed that the CN value was similar to those of forest evergreen (FRSE) and forest mixed (FRST) values in the SWAT crop database respectively.
* * *
a) An expanded description of the SWAT calibration and validation procedures is needed, which again should be in a separate subsection. This should include a description of the calibration parameters used in the study, including the default value (or initial value range) and the final calibrated values. Please also provide a description of any sensitivity analyses that was performed and provide a description of the specific baseflow separation techniques that were used in the calibration process. A description of measured baseflow data, or proxy baseflow data obtained via literature sources or expert opinion, is also important in relation to the use of the BFI indicator in your study.

We agree with the referee's suggestions, and have expanded the description of the SWAT calibration and validation procedures in subsection 2.2.1.

Model validation and calibration We calibrated the model using the Latin hypercube sampling approach from the Sequential Uncertainty Fitting version 2 in the SWAT Calibration and Uncertainty Procedure (SWAT‐CUP) package. First parameter ranges were determined based on minimum and maximum values allowed in SWAT. The SWAT‐CUP is an interface for auto-calibration that was developed for SWAT. The interface links any calibration/uncertainty or sensitivity program to SWAT (Abbaspour, 2015). The discharge data of BH and MT watersheds used for calibration and validation were available for the period of 2005-2014. The calibration was carried out in year 2007-2009 and the validation in year 2012-2014. We evaluated the model using Nash-Sutcliff efficiency (NSE) and Percent Bias (PBIAS). The NSE is a normalized statistic that determines the relative magnitude of the residual variance ("noise") compared to the measured data variance ("information") (Nash and Sutcliffe, 1970). The PBIAS measures the average tendency of the simulated data to be larger or smaller than the

observations Gupta et al., (1999). The optimum value is zero, and low magnitude values indicate better simulations. Positive values of PBIAS indicate model underestimation and negative values indicate model overestimation. The model input parameters that were used for the calibration process and their fitted values after calibration are shown in Table 4.

The ALPHA_BF (baseflow recession constant) was calculated from daily streamflow hydrograph plotted on semi-log paper. The calculation was based on the average ALPHA_BF values derived from several selected individual rainfall events. Based on several study in Indonesia, the ALPHA-BF ranges from 0.9 to 0.95.

⎯⎯⎯⎯⎯⎯⎯⎯⎯⎯⎯⎯⎯⎯⎯⎯⎯⎯⎯⎯⎯⎯⎯

b) I suggest you then introduce a third subsection that describes the specific C and BFI methods that were used in your analyses.

We agree with the referee's suggestions, and have added subsection 2.2.2 to describes the specific C and BFI methods.

2.2 Flow simulation

2.2.1 SWAT model

2.2.2 Simulated C and BFI values

SWAT model simulated daily flow components including total stream flow, surface flow and base flow in all 48 sub-watersheds of BH and MT watersheds (Figure 7). We calculate the daily average ratio of the surface flow to the rainfall and the baseflow to the total streamflow as a proxy to C value (direct runoff coefficient) and BFI value (baseflow index) for each sub-watershed.

⎯⎯⎯⎯⎯⎯⎯⎯⎯⎯⎯⎯⎯⎯⎯⎯⎯⎯⎯⎯⎯⎯⎯

13) Please expand on your discussion of the calibration and validation results. This should include showing hydrograph comparisons between the simulated and measured

outputs and discussion of your results in the context of model evaluation criteria suggested in the two Moriasi et al. studies.

We agree with the referee's suggestions. We have expanded our discussion of the calibration and validation in a new subsection (Subsection 3.1 – Simulated flow) including NSE and PBIAS as suggested in the two Moriasi et al. (2007, 2012) Moriasi et al. (2007, 2012) recommend three quantitative statistics, Nash-Sutcliffe efficiency (NSE), percent bias (PBIAS), and ratio of the root mean square error to the standard deviation of measured data (RSR) be used in model evaluation.

A visual comparison between the best-fit simulations and observed data is depicted in Figure 8. Overall, the model performance was satisfactory with the Nash-Sutcliff efficiency values of 0.80-0.88, (calibration) and 0.84 - 0.85, (validation); and the PBIAS values of -2.9 - 1.2, (calibration) and 7.0-11.9 (validation) for the BH and MT watersheds respectively.
* * *
14) Line 131: I think the word "was" should be "were". Why were simulated values that were within an "order of magnitude" of the measured values considered acceptable? It appears that the average measured and simulated C values reported in Tables 5 versus 6 were almost identical; that would indicate that the "order of magnitude" criteria is unnecessary?

We thank the referee for pointing this out. We have revised the entire sentence:

Line 131-132 In addition to the SWAT model calibration and validation procedure, we also compared the simulated C and BFI values with those obtained from the field measurement in selected watersheds.
* * *
15) Sentence in lines 184-185: The phrase "as acceptable for a good watershed service" in this sentence sounds odd. A suggested revision is: "The Ministry of Forestry

of Indonesia considers C values < 0.35 to be adequate to support required ecosystem services for Indonesian watersheds (citation)."

We agree with the referee's suggestions, and have revised the sentence accordingly.

Line 184-185 The Ministry of Forestry of Indonesia considers C values < 0.35 to be adequate to support required ecosystem services for Indonesian watersheds (Ministry of Forestry Decree, 2013)

———————————————————————————

16) Conclusions: Some expansion of your Conclusions section is warranted. Please include additional quantitative information from both the baseline testing results as well as the C and BFI analyses.

We agree with the referee's suggestions and have included additional quantitative information in the conclusion.

Line 230-239 Overall, the SWAT model performance was satisfactory with the Nash-Sutcliff efficiency values of 0.80-0.88, (calibration) and 0.80 - 0.85, (validation); and the PBIAS values of -2.9 - 1.2, (calibration) and 7.0-11.9 (validation). The study exhibits a statistically significant correlation of percentage of forest covers in a watershed with C (negatively) and BFI (positively). On the other hand, the rubber and oil palm plantations showed flow regulation behavior contrary to forest covers. Finally the study suggests the minimum forest cover requirement in the study area (i.e. 30%) for sustainable ecosystem services. The quantitative relation between forest cover and flow indicators derived in this study can be used as a guide for the regional planners to determine the minimum proportion of forest conservation area to maintain a sustainable ecosystem service of water flow regulation in a watershed.

———————————————————————

[Figure]

Figure 1. Bulk density in the study area under different land use types. Different letters indicate significant differences of averages according to a Bonferroni-corrected posthoc t-test based on an ANOVA (p< 0.05)

**Fig. 1.**

Figure 2. Study area in the Jambi Province, Sumatra Island, Indonesia (a) macro watershed (b) small watershed (c).

**Fig. 2.**

[Figure]

Figure 3. Soil type in BH and MT watersheds. Soil types respresent Fluvaquents (S-3), Humitropepts (S-4), Paleudults (S-7), Tropofluvents (S-10), Troposaprist (S-11), Tropodults (S-12), Dystrandepts (S-13), Dystropepts (S-14).

**Fig. 3.**

Figure 4. Oil palm hydrological components

**Fig. 4.**

[Figure]

Figure 5. Interception of different plantation crops and forest. Different letters
indicate significant differences of averages according to a Bonferroni-corrected
posthoc t-test based on an ANOVA ($p < 0.05$)

**Fig. 5.**

[Figure]

Figure 6. Surface runoff of different land use types in the study area. The different letters indicate significant differences among averages according to a Bonferroni-corrected posthoc t-test based on an ANOVA ($p< 0.05$).

**Fig. 6.**

Figure 7. Land-use types and the sub-watershed numbering of the BH (a) and MT (b) watersheds. The box whisker plots (c and d) represent the distribution of the proportion of the land-use types in all sub-watersheds in both watersheds.

**Fig. 7.**

[Figure]

[Figure]

Figure 8. Observed vs. simulated streamflow and 95% uncertainty interval (95PPU) of behavioral simulations over time (defined as simulations with Nash-Sutcliff efficiency NSE > 0.5) for the BH (a) and MT (b) watersheds respectively.

**Fig. 8.**

Table 1. Model input data sources for the watershed modeling

| Data type | Resolution | Description | Source Data |
|---|---|---|---|
| Topography | 30 m | Digital Elevation Model with 30 m pixel resolution | LAPAN |
| Soils | 1:250,000 | Soil hydraulic conductivity, bulk density, available water content and texture were resampled in the field | Soil Research Institute, Ministry of Agriculture |
| Land use | 1:100,000 | Land use map with intensive ground check | Regional Planning office (BAPPEDA) |
| Rainfall and climate | Daily | Rainfall stations (Rantau Pandan, Siulak Deras, Muara Imat); climate station (Jambi, Pematang Kabau and Bungku) | BMG office (Meteorology and Geophysics Agency) and CRC990 |
| Streamflow | Daily discharge data | Stations: Muara Tembesi, Rantau Pandan, Air Gemuruh, Batang Tabir, Batang Pelepat, Muara Kilis | Ministry for Public work (BBWS) |

LAPAN: National Aeronautics and Space Agency (*Lembaga Antarikasa dan Penerbangan Nasional*), BAPPEDA: Regional Planning Agency (*Badan Perencanaan Daerah*), BMG: Meteorology and Geophysics Agency (*Badan Meteorologi dan Geofisika*), CRC990: Collaborative Research Centre 990, BBWS: Catchment Regional Agency (*Balai Besar Wilayah Sungai*)

**Fig. 9.**

Table 2. Adapted input of crop-parameters different land-use types

| SWAT input parameters | | Oil palm | Rubber | Agro-forest | Forest |
|---|---|---|---|---|---|
| CANMX | Original | 0 | 0 | 0 | 0 |
| | Adapted | 4.0 | 2.7 | 4.3 | 5.8 |
| OV_N* | Original | 0.14 | 0.14 | 0.1 | 0.1 |
| | Adapted | 0.07 | 0.14 | 0.4 | 0.5 |
| BLAI | Original | 5 | 2.6 | 5 | 5 |
| | Adapted | 3.6** | - | - | - |
| CHTMX | Original | 3.5 | 3.5 | 6 | 10 |
| | Adapted | 12 | | | |
| T_BASE | Original | 7 | 7 | 0 | 10 |
| | Adapted | 20 | 20 | 20 | 20 |
| T_OPT | Original | 20 | 20 | 30 | 30 |
| | Adapted | 28 | 28 | 30 | 30 |

\* OV_N values are low in oil palm and rubber due to the clean weeded management practice
\*\* Meijide et al. (2017)

**Fig. 10.**

Table 3. Adapted CN values for typical land use types in the study area

| Landuse | | Hydrologic Soil Group | | | |
|---|---|---|---|---|---|
| | | A | B | C | D |
| Oil palm | Original CN value | 45 | 66 | 77 | 83 |
| | Adapted CN value | - | - | - | **83** |
| Rubber | Original CN value | 45 | 66 | 77 | 83 |
| | Adapted CN value | - | - | - | **83** |
| Agroforest (FRST) | Original CN value | 36 | 60 | 65 | 79 |
| Forest | Original CN value | 25 | 45 | 70 | 77 |

**Fig. 11.**

Table 4. The calibration parameters used in the study, including the initial value range and the final calibrated values for the BH and the MT watersheds respectively.

| Parameters | Descriptions | Initial value range | Best fit values | |
|---|---|---|---|---|
| | | BH and MT | BH | MT |
| ALPHA_BF | Baseflow recession constant | 0.0 – 1.0 | 0.94 | 0.91 |
| GW_DELAY | Groundwater delay time (days) | 30 - 450 | 62.5 | 57.2 |
| GWQMN | Water depth in a shallow aquifer for a return flow (mm H2O) | 0.0 – 2.0 | 0.99 | 0.45 |
| GW_REVAP | Evaporation from the ground water (mm) | 0.0 - 0.02 | 0.13 | 0.07 |
| CH_N2 | Manning's "n" value for the main channel | 0.0 – 0.3 | 0.05 | 0.15 |
| CH_K2 | Eff. hydraulic conductivity in the main channel alluvium (mm/hr) | 5.0 - 130 | 35.6 | 24.4 |
| SOL_AWC | Available water capacity of the soil (mm H2O/mm soil) | - 0.2 – 0.4 (V)[a] | 0.09 | 0.04 |
| SOL_K | Saturated hydraulic conductivity (mm h$^{-1}$) | - 0.8 – 0.8 (V)[a] | 0.71 | 0.12 |
| OV_N | Manning's "n" value for overland flow | - 0.2 – 1.0 (V)[a] | 0.51 | 0.29 |

[a] (V) = Variable depending on land-use and soil, changes in calibration were therefore expressed as fraction

**Fig. 12.**

---

## Author Response (AR1)

The manuscript deals with the effect of land cover and/or land use on a watershed response functioning. The authors investigated the influence of forest and monoculture plantations (oil palm and rubber plantations) on rainfall partitioning to direct runoff and subsurface flow for a humid tropical watershed in Indonesia. The results are based on streamflow as simulated by a calibrated SWAT model and observations across several watersheds and subsequently derived the direct runoff coefficient (C) and the baseflow index (BFI). The study exhibits a statistically significant correlation of percentage of forest covers in a watershed with C (negatively) and BFI (positively). On the other hand, the rubber and oil palm plantations showed flow regulation behavior contrary to forest covers. Finally the study suggests the minimum forest cover requirement in the study area (i.e. 30%) for sustainable ecosystem services. The topic is of current scientific interest and several studies have also investigated previously.

However, the manuscript requires a substantial improvement of the methodology and, results and discussion to be publishable. Furthermore, the manuscript would benefit a lot with the inclusion of more discussions in the introduction section from previous similar studies in the tropical regions.

General comments

1. Given the previous several studies on the effect of land cover/use conversion on the hydrology of a watershed, the introductory section needs further literature review in this regard. It should also highlight the new contribution of this manuscript.

We appreciate the referee's suggestions, and have added more literature review on the effect of land cover/use conversion on the hydrology of a watershed in the introductory section. (Line 104-114) and highlight the new contribution of this manuscript (Line 119-124)

2. I think the organization of the methods section, in general, requires restructuring and further information. For example, there is no section that describes the general SWAT model and the SWAT model for the study area, which are important for general readers and non-SWAT users.

We agree with the referee's suggestions, and have re-structured the method section as follows:

The general SWAT model is described in Line 163- 182 and the SWAT model for the study area in Line 183-282

3. Section 3.2 and section 3.3 should be presented before section 3.1. Logically thinking, observation based model evaluations should be presented first and then results of analyses based on the model simulation.

We agree with the referee's suggestions, and have re-structured the discussion section as follows:

4. No information is provided in the manuscript about the SWAT parameters, particularly the ones that control the surface runoff and the baseflow process. I think information about some of the sensitive parameters would give a good discussion points on the flow regulation behavior of different Landover/use in the study area. What was your observation on the calibrated SWAT parameters such as CN2, SOL_AWC, ALPHA_BF and CANMX among other?

We agree with the referee's suggestions, and have added detail information about mentioned SWAT parameters (Line 189-257).

5. The calibration and validation strategy are not clearly stated, albeit its importance in interpreting simulation outputs from SWAT. The calibration and validation period need to be explicitly stated. Which automatic calibration algorithm was used in SWAT-CUP? It is also essential use multiple evaluation criteria.

We agree with the referee's suggestions, and have described in detail the calibration and validation strategy and period (Line 283-305)

6. I encourage the authors to explicitly discuss the SWAT model simulation results are mainly arising due to changes in land cover not by wrong parameterization. SWAT is a highly parameterized model, therefore we might get the expected patterns for the wrong reason. This could be addressed by referring the calibrated SWAT parameters.

We appreciate very much the referee's concerns, and have explicitly discussed the SWAT model simulation to ensure that the results are mainly arising due to changes in land cover not by wrong parameterization. (Line 176-262).

Specific comments

1) Lines 1-2: I suggest to check the title. i) Since it is an application in tropical region in Indonesia, it needs to be specific. ii) It seems to me some action words are missing. You could simply add, for instance, "requirement" that reads as "Minimum forest cover requirement for sustainable water flow regulation: A case study in a watershed under rapid expansion of oil palm and rubber plantations in Indonesia"

We appreciate the referee's suggestion. The title has been adjusted.

2) Lines 9-32: The abstract could be shortened to a certain extent by reducing the seemingly redundant sentences on flow regulation functioning and benefits, keep the most important points only.

We agree with the referee's suggestions, and have improved the abstract accordingly (Line 11-34).

3) Lines 14-15: It is a bit confusing sentence, please improve the language.

The referee appears to be correct. We have removed the confusing sentence from the abstract.

4) Line 40 " Lele, 200" please add 0 (

We thank the referee for the correction (Line 76)

5) Line 40 "Functional water flow regulation reduces flood peaks by moderating direct runoff." It would be nice to add some references here.

We have provided the relevant reference (Line 77)

6) Line: 46:"base flow" remove space

Revision made; we inserted the space (Line 80)

7) Line 46: ")]" remove the square bracket

Revision made; we removed the square bracket (Line 80)

8) Lines: 69-71: Please improve the language

We thank the referee for pointing this out; we have improved the language Line 70-72.

9) Lines 72-73: Improve the language, for instance, "Distributed hydrologic models such as the Soil and Water Assessment Tool (SWAT) are useful to understand the effects of land use changes on watershed flow regulation: : :: : ::"

We thank the referee for pointing this out; we adapted the sentences suggested by the referees. (Line 97-98)

10) Lines 104-105: ": : :: : : is the direct runoff ratio of to rainfall." should be "is the ratio of direct runoff to rainfall"

We thank the referee for the correction. We have revised the sentence ( Lines 164)

11) Line 88: Please add the size of the study area and perhaps the location coordinates.

We have added the size of the study area (Line 140) and the location coordinates (Line 127).

12) Lines 88-93: It would be informative to add information on the historical land cover change in the study area.

We thank the referee for the suggestion. We have added more information on the historical land cover change in the study area. (Line 128-137).

13) Lines 94-98. I think the methodology description should not be included in study site description. I suggest to move this part to appropriate subsection in the methodology.

We thank the referee for the suggestion. We have removed this part.

14) Line 99: Replace "&" with and

Revision made;  we replaced "&" with "and" (Line 306)

15) Line 101: "C & BFI" it should be "C and BFI" like in the abstract section and it should be consistent throughout the manuscript.

Revision made;  we replaced "&" with "and"

16)Lines 102-104: Please improve the language. And it is somewhat similar with Lines 109-110

We thank the referee for pointing this out; we have removed the duplication.

17)Lines 104-109: This is confusing! This describes the general SWAT model and I would rather expect a separate subsection for it. This should also tell how SWAT computes surface runoff, baseflow: : :...See the comments in the general comment.

We thank the referee for the suggestion. We have added subsection 2.2 under which we have described the general SWAT model  and the model setup  (Line 163- 282)

18) Line 114: I would prefer the areas in km2.

Revision made; we have replaced "ha" with "km$^2$" (Line 140)

19)Line 118-121: Describes the SWAT model setup for the study area. Therefore, I would expect to get this information before describing section 2.2 (Simulated C and BFI) values.

We thank the referee for the suggestion. We have described the SWAT model setup in sub-section 2.2.1 (Line 183)

20)Line 122: Add SWAT-CUP reference

Revision made; we have added the SWAT-CUP reference (Line 289-290)

21)Lines 121-129: This part tries to elaborate the model calibration and evaluation part. SWAT-CUP provides several options for model calibration, which one did you use in this study? Please be specific. When is your calibration and validation periods? I suggest separate subsection for model calibration and evaluation approach.

We thank the referee for pointing this out. We have elaborated the model calibration and validation in subsection 2.2.2 (Line 283).

22)Line 128: As demonstrated in several studies, NSE is sensitive to peak flows. You calibrated and evaluated your model using only NSE. How do you justify this? I think it would be good to add a few more performance indices in the evaluation so that the reader would have a better feel on the reliability of the model simulation outputs.

We thank the referee for the suggestion. We have added one more indices, i.e. Percent bias (PBIAS) for the evaluation. Percent bias measures the average tendency of the simulated data to be larger or smaller than the observations. The optimum value is zero, where low magnitude values indicate better simulations. Positive values of

PBIAS indicate model underestimation and negative values indicate model over estimation (Line 296-305)

23)

Correction made

24) Line 158 "didn't" should be "did not"

Correction made

25) Lines 162-163 repetition see line 121

We thank the referee for pointing this out. We have removed the repetition

26) Line 163-164: Add more statistics

Revision made; we have added more statistics, namely percent bias (PBIAS). The PBIAS measures the average tendency of the simulated data to be larger or smaller than the observations. The optimum value is zero, and low magnitude values indicate better simulations. Positive values of PBIAS indicate model underestimation and negative values indicate model overestimation. (Line 296-305)

27) Lines 165-167: what did you obtain from the comparison? How much they agree? What statistical measures did you use?

We thank the referee for pointing this out. We explained the comparison in Line 333-336. It was just a simple arithmetic comparison.

28) Lines 168-173: More suitable in the methodology section.

We agree with the referee suggestion. We have moved the sentences to the methods section (Line 310-314)

29) Lines 180-184 Too long sentence, it is better to follow simple sentences. Improve the language as well.

Revision made. We have improved the language (Line 367-369)

30) Line 182: Oil palm harvest and oil palm circle are equal (i.e. 3 cm h-1).

Revision made. We have improved the language

31) Lines 185-188: I'm puzzled by this conclusion. Is the rainfall distribution similar throughout the basin? Because if there is a spatial variation in rainfall magnitude, the effects of forest conversion on the flow regulation would vary accordingly.

We thank the referee for raising this question. We agree that there is always spatial variation in rainfall magnitude throughout the basin from one event to another event. Both watersheds in our study were partitioned into 48 sub-watershed, which reduce the degree of rainfall spatial variability in the watersheds.

32) In Figure 4a, I see a C value less than 0.35 for forest cover about 20%, what do you think about this?

We thank the referee for raising this question. We have explained the reason in Line 370-373.

33) Line 207: please improve the language

We have omitted this particular paragraph due to the lack of field sample data to construct the graph statistically sufficient.

34) Lines 207-214. I think this need more discussion. SWAT has a known limitations in simulating the low flow regime and that would have an effect on the BFI, as also mentioned by the authors. See the recent study for further discussion: Pfannerstill, M., B. Guse, and N. Fohrer, 2014a. A Multi-Storage Groundwater Concept for the SWAT Model to Emphasize Non- linear Groundwater Dynamics in Lowland Catchments. Hydro- logical Processes 28:5599-5612, DOI: 10.1002/hyp.10062

We agree with the referee, the SWAT version used in this study has limitation to model the groundwater component of the streamflow. We have enriched the discussion on BFI with the suggested literature (Line 395-401).

35) Line 344: ": : :MT(b,: : :"

Revision made. We thank you the referee for the correction

36) Line 376: Table 3, In MT watershed sub.wat.nr 23 has a 100% forest cover but the BFI is low, meaning low baseflow contribution from the groundwater. Justify this in the discussion.

We thank the referee for raising this question. We have explained the reason in Line 373-377

37) Line 379 Table 4, Please recheck the numbers and the calculations.

We have re-checked and corrected the errors. The total average of the C values of 0.59 remains unaffected.

Anonymous Referee #2

General comments:
I found the topic and results described in this manuscript to be quite interesting. There is very limited information available in the literature to date regarding the potential effects of expanded production of rubber or oil palm trees, using SWAT model or any other modeling approach. Thus I think that the information reported in this manuscript will ultimately prove to be a useful contribution to Hydrology and Earth System Sciences (HESS) and the existing literature in general. However, I believe that the current manuscript suffers from several deficiencies including inadequate review of existing literature, insufficient description of SWAT and key input parameters (including coefficients used for rubber tree and oil palm tree in the crop parameter file), lack of in-depth description of SWAT calibration and validation results, and an inadequate description of the simulated watersheds. Specific comments regarding these issues are provided below.

We appreciate the referee's concerns. We have addressed all referees' concern in the respective comments below including: a) more comprehensive review of existing literature, b) in depth description of key crop parameters and c) adequate description of the simulated watersheds

Specific comments

1) Abstract: The Abstract needs to be considerably revised to reflect more of the actual quantitative results of the study versus the "general discussion" that dominates much of the abstract between lines 9 to 24. The revised abstract should include a summary of the baseline calibration and validation results.

We agree with the referee's suggestions. We have shortened the "general discussion". We have also included   more quantitative results (Line 28) and summary of the baseline calibration and validation results (Line 23-25).

2) Lines 43-45: I would suggest you rewrite this sentence to read something like: "This vertical movement of water in the soil determines how much water flows as direct runoff and how much percolates to the water table where it sustains baseflow or groundwater (references)."

We agree with the referee's suggestions, and have revised the sentence (Line 79-81).

3) Lines 49-68: Please include citation and discussion of some "big picture" studies regarding the impacts of Palm Oil and/or Rubber Trees in the southeast Asia region such as those listed immediately below.

We thank the referee for this suggestion. We have substantially enriched the citation and discussion of some studies regarding the impacts of palm oil and/or rubber trees in the Southeast Asia region (Line 46-64).

4) Lines 69-70: Please expand this discussion to provide a broader review of different modeling and other analysis methods, beyond the option of SWAT, available to assess the impacts of expanded rubber and oil palm plantations in the Southeast Asia region.

We thank the referee for this suggestion, and have expanded the discussion to provide a broader review of different modeling and other analysis methods, beyond the option of SWAT (Line 84-96).

5) The expanded paragraph noted in comment 3 should be followed by a specific paragraph about SWAT including relevant review studies about SWAT and a more in-depth review of how SWAT has been used for land use change analyses. Note that the Zhang et al. (2013) article you cite in line 76 is not a very good choice regarding reviews of SWAT studies; please instead cite one or more of the studies listed on the webpage at http://swat.tamu.edu/publications/special-issues/ or in the "SWAT Publications box" in http://swat.tamu.edu/. Please also cite some relevant SWAT "land use change studies" (see the SWAT Literature Database that can again be accessed on the SWAT model homepage) such as those listed here:

Babel, M.S., B. Shrestha and S.R. Perret. 2011. Hydrological impact of biofuel production: A case study of the Khlong Phlo Watershed in Thailand. Agricultural Water Management. 101(1): 8-26. DOI: 10.1016/j.agwat.2011.08.019.
Marhaento et al. 2017. Attribution of changes in the water balance of a tropical catchment to land use change using the SWAT model. Hydrological Processes. 31(11):2029–2040. DOI: 10.1002/hyp.11167.
Tan et al. 2015. Impacts of land-use and climate variability on hydrological components in the Johor River basin, Malaysia. Hydrological Sciences Journal. 60(5): 873-889. DOI: 10.1080/02626667.2014.967246.
Tarigan et al. 2016. Mitigation options for improving the ecosystem function of water flow regulation in a watershed with rapid expansion of oil palm plantations. Sustainability of Water Quality and Ecology . 8: 4-13. DOI: 10.1016/j.swaqe.2016.05.001.
Wangpimool et al. 2017. The impact of Para rubber expansion on streamflow and other water balance components of the Nam Loei River Basin, Thailand. Water. 9(1) DOI: 10.3390/w9010001.

We thank you the referee for suggestions. We have added specific paragraph about SWAT and a more in-depth review of how SWAT has been used for land use change analyses (Line 103-115).

6) Lines 71-73: These two current sentences have grammatical problems. As a part of comment 4, I suggest that you revise the text as follows: "A useful tool to answer this question is the Soil and Water Assessment Tool (SWAT) ecohydrological model (Arnold et al., 1998; 2012), which quantifies the water balance of a watershed on a daily basis (Neitsch et al., 2009) and has been recommended for the evaluation of hydrological ecosystem services of a watershed (Vigerstol et al., 2011)."

We agree with the referee's suggestions, and have improved the sentences accordingly (97-103).

7) Study area description: The two study watersheds should be described in depth in this subsection rather than being referenced later in subsection 2.2 (please describe the area of the watersheds in km2 rather than ha). More detailed land use information (percentages of each type of land use) for the two watersheds should be provided (rather than waiting until subsections 2.3.1 and 3.2 to describe some of that information), as well as more information about the natural vegetation, and rubber and oil palm plantations (growth cycles, management practices, time period of plantation development, etc.). Further details about the typical porosity and other characteristics of the soils in the study watersheds would also be useful.

We thank the referee for the suggestion. We have re-restructured and substantially improved the description of the whole subsections in the method section. We have described in depth the study watersheds and land use information (Line 127-148), more information about the natural vegetation, and rubber and oil palm plantations (Line 152-162) and characteristics of the soils (Line 142-143).

8) In relation to comment 7, some description of all six macro watersheds shown in Figure 1 should also be provided in the Study area description subsection. Who defined these six watersheds and why? It is clear that hydrologic data was collected for the watersheds but the current text is vague regarding the overall purpose of these six watersheds.

We included only 2 macro watersheds and two small watersheds in the analysis. The other four watersheds, which were previously used to analyze observed BFI value in a watershed scale, were excluded. The reason was that these four watersheds were considered insufficient statistically to represent the observed BFI value for the whole study area.

9) Also in relation to comment 7, please describe the "small watersheds" referenced in lines 144-145 and 195-196 and shown in Figure 1 in the study area subsection, rather than waiting to describe those in current section 2.3.1

(and that information does not need to be repeated at the start of section 3.2). What other hydrologic data were collected for those small watersheds besides the C values?

We agree with the referee's suggestions, and have described "small watersheds" in Line 144-148. Additional description of small watershed are provided in the methodology section of measured C values (Line 322-329).

10) Please rewrite "C&BFI" as "C and BFI" throughout the text.

Revision made

11) A SWAT Description subsection needs to be added to the manuscript. This should note the specific version of the model used for the study (including the Revision number) and provide a succinct overview of the model, especially regarding components that were particularly important for the study you conducted. A description of the crop parameters used for the rubber and oil palm trees, and other vegetation in the watersheds, should also be provided (those parameters could be described later in the methods if more appropriate). See the Wangpimool et al. article listed in comment 4 above regarding revised rubber tree crop parameters they used in their study.

We agree with the referee's suggestions, and have described the SWAT model and crop parameter in subsection 2.2 (Line 163-257).

a) An expanded description of the SWAT calibration and validation procedures is needed, which again should be in a separate subsection. This should include a description of the calibration parameters used in the study, including the default value (or initial value range) and the final calibrated values. Please also provide a description of any sensitivity analyses that was performed and provide a description of the specific baseflow separation techniques that were used in the calibration process. A description of measured baseflow data, or proxy baseflow data obtained via literature sources or expert opinion, is also important in relation to the use of the BFI indicator in your study.

We agree with the referee's suggestions, and have expanded the description of the SWAT calibration and validation procedures in the Section 2.2.2 (Line 283-.305). The default value (or initial value range) and the final calibrated values has been described in Line 350-352. The baseflow separation technique was described in Line 359-262.

b) I suggest you then introduce a third subsection that describes the specific C and BFI methods that were used in your analyses.

We agree with the referee's suggestions, and have added subsection 2.3 to describes the simulated C and BFI methods (Line 306-320).

13) Please expand on your discussion of the calibration and validation results. This should include showing hydrograph comparisons between the simulated and measured outputs and discussion of your results in the context of model evaluation criteria suggested in the two Moriasi et al. studies.

We agree with the referee's suggestions. We have expanded our discussion of the calibration and validation results including NSE and PBIAS as suggested in the two Moriasi et al. (2007, 2012). Moriasi et al. (2007, 2012) recommend three quantitative statistics, Nash-Sutcliffe efficiency (NSE), percent bias (PBIAS), and ratio of the root mean square error to the standard deviation of measured data (RSR) be used in model evaluation (Line 345-361)

14) Line 131: I think the word "was" should be "were". Why were simulated values that were within an "order of magnitude" of the measured values considered acceptable? It appears that the average measured and simulated C values reported in Tables 5 versus 6 were almost identical; that would indicate that the "order of magnitude" criteria is unnecessary?

We thank the referee for pointing this out. We have revised the entire sentence (Line 334-336).

15) Sentence in lines 184-185: The phrase "as acceptable for a good watershed service" in this sentence sounds odd. A suggested revision is: "The Ministry of Forestry of Indonesia considers C values < 0.35 to be adequate to support required ecosystem services for Indonesian watersheds (citation)."

We agree with the referee's suggestions, and have revised the sentence accordingly (Line 383-387).

16) Conclusions: Some expansion of your Conclusions section is warranted. Please include additional quantitative information from both the baseline testing results as well as the C and BFI analyses.

We agree with the referee's suggestions and have included additional quantitative information in the conclusion (Line 426-432).

---

## Referee Report (RR1)

[referee-annotated manuscript omitted]